# Laser-Induced Breakdown Spectroscopy Applied to Elemental Analysis of Aqueous Solutions—A Comprehensive Review

Nils Schlatter *[ID] and Bernd G. Lottermoser [ID]

Institute of Mineral Resources Engineering, RWTH Aachen University, Wüllnerstraße 2, 52062 Aachen, Germany; lottermoser@mre.rwth-aachen.de

* Correspondence: schlatter@mre.rwth-aachen.de

**Abstract:** Laser-induced breakdown spectroscopy (LIBS) has evolved considerably in recent years, particularly the application of portable devices for the elemental analysis of solids in the field. However, aqueous analysis using LIBS instruments, either in the laboratory or in the field, is rather rare, despite extensive research on the topic since 1984. Thus, our comprehensive review aims to provide a clear overview of this research to offer guidance to new users. To achieve this, we examined the literature published between 1984 and 2023, comparing various settings and parameters in a database. There are four different categories of LIBS instruments: laboratory-based, online, portable, and telescopic. Additionally, there are four main categories of sample preparation techniques: liquid bulk, liquid-to-solid conversion, liquid-to-aerosol conversion, and hydride generation. Various experimental setups are also in use, such as double-pulse. Moreover, different acquisition settings significantly influence the sensitivity and therefore the detection limits. Documentation of the different methods of sample preparation and experimental settings, along with their main advantages and disadvantages, can help new users make an informed choice for a particular desired application. In addition, the presentation of median detection limits per element in a periodic table of elements highlights possible research gaps and future research opportunities by showing which elements are rarely or not analysed and for which new approaches in sample preparation are required to lower the detection limits.

**Keywords:** laser-induced breakdown spectroscopy (LIBS); elemental analysis; aqueous solutions; trace elements; environmental analysis; detection limits; liquid sample preparation techniques; signal enhancement

## 1. Introduction

The analysis of aqueous solutions for different elements is important for numerous reasons, such as environmental analysis [1], resource estimation [2], hydrogeochemical research questions [3], process control in nuclear fuel reprocessing plants [4], industrial control [5–7], food analysis [8], and medical applications [9]. Today, elemental analysis of aqueous solutions is typically performed in the laboratory using well-established methods such as ion chromatography (IC), atomic absorption spectroscopy (AAS), inductively coupled plasma atomic emission spectroscopy (ICP-AES), and inductively coupled plasma mass spectrometry ICP-MS [10,11].

Another possible analytical technique that is capable of analysing aqueous solutions is laser-induced breakdown spectroscopy (LIBS). The atomic emission spectroscopy technique was developed shortly after the introduction of the first lasers in the 1960s [12]. It involves a low-energy, pulsed laser focused on a sample by a lens. A small volume of the sample is vaporised in a plasma which is created by the high energy density at the surface, and the emitted light is collected and sent to a spectrometer where it is dispersed [13]. With the use of a detector, the signals are recorded and digitised. The spectrum can be evaluated with suitable software and the atoms, ions and simple molecules contained in the plasma

can be determined qualitatively via the characteristic lines [13]. By using certified reference materials (CRMs), it is possible to establish a calibration that also quantifies the elements and molecules present. The advantages of LIBS are typically minimal sample preparation, low instrumentation cost, rapid analysis, simultaneous detection of multiple elements, and the possibility of in situ analysis, real-time analysis, and remote analysis [14,15]. In particular, the analysis of solids, including soils, geological samples, archaeological samples, metals, and alloys using portable LIBS (pLIBS) devices has developed enormously in the last few years [16] and the applications are hardly limited. As a result, pLIBS has established itself as a powerful competitor and companion to portable X-ray fluorescence (pXRF) in field analysis of solids, as it is theoretically capable of analysing the entire periodic table of elements [16,17].

However, the analysis of aqueous solutions with LIBS, whether in the laboratory or in the field, has since played a rather niche role among potent and widely used analytical methods such as ICP-MS, ICP-AES, AAS, and IC [11]. This may be due to the complex interactions during the formation of the plasma at the surface of the liquid [10], or simply because the other methods have become so well-established that it has been difficult for LIBS to establish itself.

In the meantime, numerous sample preparation techniques and experimental setups that circumvent the problems mentioned have been developed and presented in the literature [15,18]. These enable detection limits for several elements that compete with ICP-MS. Figure 1 illustrates the constant increase in publications on the topic of aqueous solution analysis by LIBS since the first application in 1984 by Cremers et al. [19]. Harun and Zainal already gave a comprehensive overview of different sample preparation techniques in liquid analysis [15]. However, the diversity and variety of different instrument types, sample preparation techniques, LIBS experimental setups, and acquisition settings can be confusing to new users. This may hinder the more widespread use of LIBS in laboratories for aqueous analysis. Therefore, the main aim of this review is to provide a comprehensive overview of our existing knowledge on the analysis of elements dissolved in aqueous solutions using LIBS and to provide an initial guide for new LIBS users.

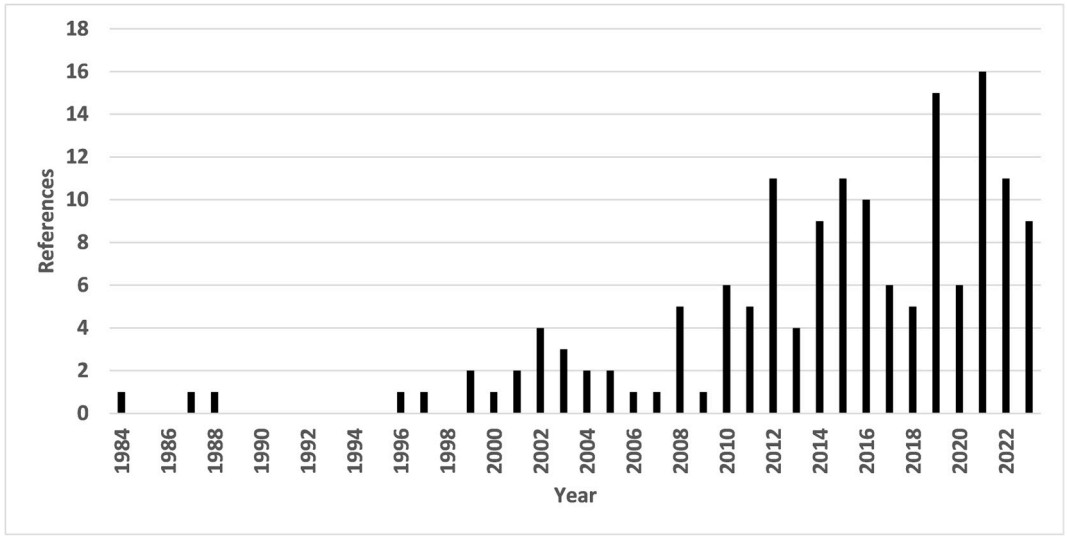

**Figure 1.** Number of references reporting one or more detection limits for the analysis of aqueous solutions by LIBS from 1984 to date. Data based on the literature reviewed.

Firstly, historical and spatial aspects of research into elemental analysis in aqueous solutions using LIBS are discussed. Sample types and sample preparation techniques to overcome the physical issues are then elaborated on. It is highlighted which sample preparations have given surpassing results and which are not recommended. The focus is less on a comprehensive discussion of the various methods of sample preparation, but rather

on which methods are more commonly used and give better results for a specific application. The different types of instruments and their main advantages and disadvantages are also presented. For signal enhancement, different experimental setups are used and their main advantages and disadvantages are discussed. In addition, the most important acquisition settings are presented, as they strongly influence the analysis results. The analysis of the frequency of the different acquisition settings and the different sample preparation techniques used may help scientists who want to use LIBS for their research to make an informed choice for a particular application.

Another aim of this work is to demonstrate the opportunities and limitations of LIBS analysis of elements dissolved in aqueous solutions and associated research gaps. Although in theory any element can be analysed by LIBS [16], it turns out that some elements are less quantifiable than others. There is a large number of publications that provide detection limits for a wide variety of elements dissolved in aqueous solutions using LIBS. For example, the very early publication by Cremers et al. showed the possibility to analyse Li, Na, K, Rb, Cs, Be, Mg, Ca, B, and Al in aqueous solutions [19]. Since then, numerous other elements have been successfully tested and detection limits documented, such as for potentially toxic elements like Cr and Pb [20] or lanthanides [21]. The present work presents these detection limits explicitly for the analysis of aqueous solutions for as many elements as possible. The aim is to give a simple, accessible sense of which elements are commonly investigated, which are rarely studied and, more importantly, a sense of the sensitivity per element. The results of the literature review are among other things presented in this work in the form of an annotated periodic table (cf. graphical abstract), thus providing a quick overview of the analysis of aqueous solutions for a wide range of elements using LIBS. Moreover, the reviewed publications are briefly presented with the sample preparation techniques, sample types, LIBS experimental setups and acquisition settings used, and recommendations are given. Therefore, this review contributes to a better understanding of LIBS as a technique for the analysis of dissolved elements in aqueous solutions.

## 2. Methods

A comprehensive literature review was conducted using databases and search engines such as Scopus, PubMed, Google Scholar, GeoRef, Web of Science, ResearchGate, and Bielefeld Academic Search Engine (BASE) with no annual limit on the search query. However, this review only documents research published until August 2023. Typical search items were: element name, LIBS/LIPS, water analysis, quantitative analysis, environmental analysis, aqueous, aqueous solution, solution, surface enhanced, hydride, hydride generation, water, aquatic, liquid, aerosol, metal ion, cation, and anion.

All references and the references therein covering the analysis of aqueous solutions using any form of LIBS were collected. Sample types in the references included the dilution series of stock solutions; aqueous artificial solutions such as industrial or waste waters; biological solutions such as blood, urine, or wine; natural solutions such as river water, rainwater, or groundwater; and natural saline solutions such as seawater or brine. The focus of the literature review was clearly on the analysis of aqueous solutions with any form of LIBS, excluding other liquids like oils, liquid metals, melts, resins, emulsions, and colours. Also, colloidal or particulate analysis in liquids was excluded in the review. These exclusions were made to keep the matrix reasonably similar for comparison in terms of viscosity, density, and general composition, as there is a strong matrix dependence of LIBS analysis [22].

The focus of this review was not to describe in meticulous detail every possible method of sample preparation technique used with LIBS, or every LIBS configuration. Comprehensive reviews of sample preparation techniques can be found elsewhere [15,18]. Rather, it is intended to provide guidance to scientists wishing to use LIBS for the first time in their research on aqueous solutions. It is therefore intended to be a comprehensive collection of published literature on the subject of aqueous solution analysis. The purpose

is to give an idea of which elements typically work well with LIBS applied to aqueous solutions and which methods have achieved low limits of detection (LoDs).

All selected publications (from 1984 to 2023) were systematically investigated for the following settings and parameters:

- Element(s) investigated,
- LoD(s) achieved,
- Sampling method/sample pre-treatment,
- Peculiarities in LIBS setup and beam geometry,
- Laser wavelength used,
- Pulse energy used,
- Year of publication,
- First author country of affiliation,
- Sample type.

This long period, in fact the entire time since the beginning of water analysis using LIBS, was chosen to obtain a comprehensive database. To compare the performance of analytical methods quickly and easily, the limit of detection (LoD) is commonly used [23]. This indicates the lowest concentration of a particular element that can be reliably determined. Typically, the LoD is determined by a calibration line that plots the measured signal as a function of analyte concentration for standards of known concentration [23]. The current attitude towards LIBS is often that the analysis is characterised by high detection limits (parts per million (ppm) to %) [23,24]. This may often be sufficient for geochemical applications or material characterisation, but for the analysis of elements dissolved in water, the typical unit of analysis is mg/L or µg/L, equivalent to ppm or parts per billion (ppb), using the simplification that the density of water is $1 \text{ g/cm}^3$. However, the use of a special sample treatment can lead to a significant improvement in sensitivity and therefore the LoD.

By far the most commonly used calculation to determine the LoD in LIBS analysis is the 3σ-IUPAC criterion: the LoD is equal to three times the standard deviation of the background signal at the lowest solution concentration divided by the slope of the calibration line [23]. This formula is used although it is outdated and LIBS calibrations are associated with non-linearity due to self-absorption, leading to miscalculation of the LoD [23]. When the detection limits are compared independently of the method used, but for one type of instrument (LIBS), they should also give an indication of whether the elements are generally poorly or well analysed by the instrument compared to other elements. This should be true if enough detection limits are available. If certain elements have not been analysed or have only been analysed infrequently, this may, except in the case of a lack of interest, also indicate that there are difficulties in quantifying these elements with the measuring instrument. For ease of comparison, all LoDs reported in the literature reviewed were recalculated to mg/L. For further simplification, values documented in auxiliary measuring units such as ppm were recalculated to mg/L on a 1:1 basis.

To allow a quick assessment of whether an element is more or less quantifiable by LIBS in aqueous solutions, a periodic table of the elements based on the one by the National Center for Biotechnology Information was used to illustrate the detection limits achieved in the literature [25]. The median of all LoDs found per element was used for comparison instead of the mean to compensate for outliers. To give an impression of the data quality, the number of LoDs used for calculation is printed below the median LoD. Several LoD values can come from one reference if different sample preparation techniques, different LIBS configurations, or different spectral lines were used. For an even faster impression, the background of the elements is coloured according to threshold values. These were defined as high analytical performance with a LoD of less than 1 mg/L, greater than 1 but less than 5 mg/L for medium analytical performance, and higher than 5 mg/L for low analytical performance.

A colour code has also been introduced for quicker assessment of data quality in the other figures evaluating the detection limits. Here, red bars indicate a value calculated

from less than five LoDs. Dark green indicates a more reliable database with more than five LoDs used for calculation.

All data collection and analysis was done in a spreadsheet, which is available as a digital supplement. It also contains additional figures, such as a second periodic table using means rather than medians. The file is intended as a guide for researchers aiming to work with LIBS on aqueous solutions. A separate worksheet called "Inf" is provided for guidance through the document. All references reviewed and evaluated are documented in a worksheet called "references", including their titles and digital object identifiers (DOIs), if available. Illustrations of different sample preparation techniques were created with artistic liberty using Photoshop CS6, based on the descriptions and figures in the literature reviewed.

## 3. Results

### 3.1. Historical Aspects

Over 153 publications on the subject of the analysis of aqueous solutions using LIBS were identified as having been published between 1984 and August 2023 and reporting at least one LoD for at least one element. Figure 1 shows the number of references for the analysis of aqueous solutions by LIBS from 1984 to date reporting at least one LoD for at least one element. It also shows a clear and constant increase in the number of publications, suggesting that research is continuing in this area.

The first usage of LIBS for the analysis of aqueous solutions probably dates back to 1984, when Cremers et al. analysed Li, Na, K, Rb, Cs, Be, Mg, Ca, B, and Al in aqueous solutions [19]. They directed the pulsed laser directly into the liquid and monitored the spark light to receive the spectra. This early publication already showed a very low detection limit for the alkali elements (0.006–1.2 mg/L), but significantly higher ones for elements of the thirteenth group (20–80 mg/L) [19]. This trend seems to continue to this day. Also, Cremers et al. encountered the problem of direct liquid analysis with LIBS [19]. When they focused the laser from the top, bubbles disturbed the analysis. Therefore, they changed to focusing from the side or the bottom. However, the sample cell made of quartz did not resist for a long time due to the high pressure produced [19]. Since then, it has been shown in many publications that the direct bulk analysis of liquids is prone to several physical issues, such as splashing, liquid evaporation, and plasma cooling [10], leading to low sensitivity and low precision [26]. Therefore, many different sample preparation techniques have been adapted and developed to bypass these issues. For example, liquid-to-aerosol conversion (LAC) [27], liquid-to-solid conversion (LSC) [11], and liquid jets (LJ) [28] can solve the problems by using a different sampling technique.

Looking at the countries where research is being carried out on this topic, several research-intensive localities stand out. Figure 2 illustrates the number of references by country from 1985 to 2023 that report one or more detection limits for the analysis of aqueous solutions by LIBS. For simplification, the country of the work affiliation of the first author in the literature reviewed was used for classification. Most of the research on the analysis of aqueous solutions has been done in China, followed by the United States of America, India, Spain, Germany, Canada, and Brazil. LIBS was first used as a liquid analysis method in the USA in 1984 [19]. Based on the reviewed literature, outside the USA, the method was first applied and published in Germany, in 1996 [29]. This was followed by publications in Italy in 1997 [30], France in 1999 [31], and the Czech Republic in 2000 [32]. In China, where most publications on the topic currently come from, LIBS was first used to analyse aqueous solutions in 2010, based on the literature reviewed [33].

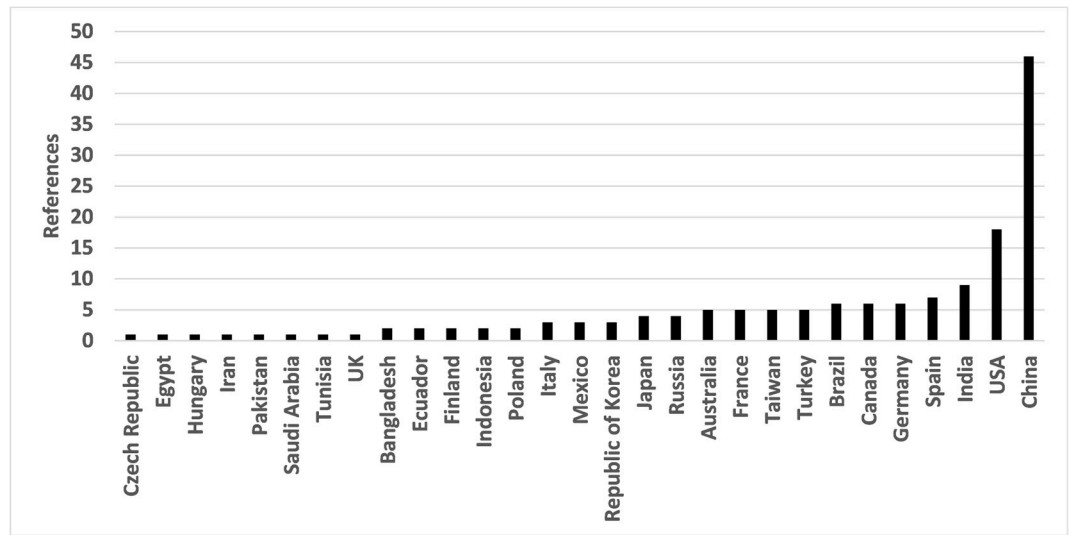

**Figure 2.** Number of references per country reporting one or more detection limits for the analysis of aqueous solutions by LIBS from 1984 to date. The affiliation of the first author is used for classification. Data based on the literature reviewed.

### 3.2. Types of Aqueous Solutions

As stated in the methods, some liquids have been excluded due to the strong matrix dependence of LIBS analysis [22]. The focus is therefore solely on elemental analysis in aqueous solutions. However, the sample types are still quite diverse due to the different applications in the literature. Table 1 shows five classes of sample types, which were used for proof of concept in the literature reviewed. For each class, examples are given and the elements analysed per class are indicated. In most cases, only dilution series of stock solutions were used, both for calibration and for proof of the developed method, even though the final application would be in a different class. Quite a few references applied their developed calibration to artificial, biological, natural, or natural saline water. Therefore, the list of elements for the stock solution is the most comprehensive one.

**Table 1.** Sample types used in the literature reviewed for calibration and verification, and elements analysed therein. Data based on the literature reviewed.

| Sample Type | Example | Elements Analysed |
|---|---|---|
| Stock solution (ss) | Prepared solutions | Li, Be, B, N, F, Na, Mg, Al, Si, P, S, Cl, K, Ca, Ti, V, Cr, Mn, Fe, Co, Ni, Cu, Zn, Ga, Ge, As, Se, Br, Rb, Sr, Y, Zr, Mo, Tc, Ru, Ag, Cd, In, Sn, Sb, Cs, Ba, La, Ce, Pr, Nd, Sm, Eu, Gd, Yb, Re, Au, Hg, Pb, Th, U |
| Artificial (art) | Industrial or waste waters | P, Cr, Ni, Cu, Zn, Ge, Cd, Au, Pb |
| Biological (biol) | Blood, urine, wine | Ti, Fe, Cu, Sr, Ag, Cs, Pb |
| Natural (nat) | River water, groundwater, rainwater | Li, B, N, Na, Mg, Al, P, S, Cl, K, Ca, Cr, Mn, Ni, Cu, Zn, As, Sr, Ag, Cd, Sn, Sb, Ba, Hg, Pb |
| Natural saline (nat sal) | Seawater, brine | Li, Na, Mg, K, Ca, Mn, Cu, Sr |

### 3.3. Sample Preparation Techniques

Direct analysis of the liquid surface by LIBS is prone to low sensitivity and repeatability [26]. This is due to effects such as evaporation, splashing, and cooling of the plasma leading to energy losses and signal intensity [10]. Therefore, many different sample preparation techniques have been developed or adapted for LIBS liquid analysis. Some use conversion processes to preconcentrate the analytes for higher sensitivity, others only to circumvent the physical issues. An overview of the most typical preparation techniques reported in the literature is given in Table 2. For illustration purposes and easier understanding, these are also shown schematically in Figure 3. Column 3 is the link between

the table and the figure, as it contains the connecting letters. Four main categories can be identified according to the aggregate state of the sample material: (1) liquid bulk (LB), (2) liquid-to-solid conversion (LSC), (3) liquid-to-aerosol conversion (LAC), and (4) hydride generation (HG) (cf. Table 2). The two main categories, LB and LSC, can be divided into several sub-categories (second column). Extensive reviews with detailed descriptions of different LIBS setups and sample preparation techniques can be found in the literature [15,18], and exemplary references are given for every category and subcategory (cf. Table 2). Therefore, this section provides only concise descriptions, divided into the four categories:

(1) LB

Surface analysis (a) can be performed by focusing the laser directly on the liquid surface [34,35]. To avoid splashing, the laser can also be focused inside the liquid (b) [1,36] or on soaked filter paper (c) [37]. Also, the analysis of low-volume isolated droplets can reduce splashing (d) [38,39]. Furthermore, a liquid jet (LJ) can be used to reduce splashing, either in a steel capillary (e) [40] or directly on the jet (f) [7,28]. Here, the laser is focused on the LJ produced by a peristaltic pump [40].

(2) LSC

An alternative method is to analyse the evaporation residue of a droplet on a substrate (g) [41,42], which should be hydrophobic to compensate for an inhomogeneous distribution. Metallic substrates are commonly used for SE analysis (h) and are prepared for homogeneous droplet distribution by geometric constraints [43] or made hydrophobic [44]. Filter-paper-supported analysis utilises the technique of evenly distributing the liquid using the filter paper (i) [45,46]. Either the dried filter paper or the evaporation residue beneath it is subsequently analysed. Also, an ion-exchange (IE) membrane can be utilised to preconcentrate the analyte (j) by either placing the membrane in the solution or filtering the solution through it [47,48]. The membrane is then dried and analysed. Moreover, the analyte can be evenly distributed on a surface through absorption by a substrate, such as wood [49], or adsorption by an adsorbent, such as activated carbon [50], which is then analysed (k). By using a chelating resin [51,52] or complexation agent, the analyte can be preconcentrated in a solid form (l). Another option is to produce a membrane by mixing the sample solution with polyvinyl alcohol (m) [53]. In nanoparticle-enhanced (NP) LSC [54,55], typically, microdroplets are dried and the particles act as an adsorbent for the analyte, cause field enhancement in the plasma, and increase the number of particles in the plasma (n) [54]. Electrospray deposition (ESD) [56,57] involves spraying a small amount of sample solution onto a heated metallic substrate using a metallic needle with a high voltage between them (o). The resulting dried residue is then analysed [57]. For electrical deposition (ED), metallic plates are placed in the sample solution, and a voltage is applied between the cathode and anode. After a few minutes of deposition, the cathode is removed and analysed (p) [58,59]. A straightforward method of LSC is freezing the liquid and analysing the surface of the resulting ice (q) [60,61].

(3) LAC

For liquid-to-aerosol generation, a nebuliser is used, and the aerosol is focused and analysed (r) [27,62].

(4) HG

To generate hydrides of certain elements, a complex system is required. This system includes a peristaltic pump, a gas–liquid separator, a flow meter, a membrane drying unit, a plasma cell, and chemicals such as acids, reductants, and a carrier gas [63,64]. The resulting hydrides can be analysed in a plasma cell [63,64].

**Table 2.** Categories of the most typical sample preparation techniques used in LIBS analysis of aqueous solutions.

| Category | Sub-Category | Cf. | Exemplary Reference |
|---|---|---|---|
| Liquid bulk (LB) | Surface | a | [34,35] |
| | Inside | b | [1,36] |
| | Soaked on filter paper | c | [37] |
| | Isolated droplet (ID) | d | [38,39] |
| | Liquid jet (LJ) capillary | e | [40] |
| | Liquid jet (LJ) | f | [7,28] |
| Liquid-to-solid conversion (LSC) | Non-surface-enhanced (n-SE) | g | [41,42] |
| | Surface-enhanced (SE) | h | [43,44] |
| | Filter-paper-supported | i | [45,46] |
| | Ion exchange (IE) | j | [47,48] |
| | Adsorption/absorption (Ad/Ab) | k | [49,50] |
| | Chelating/complexation (Che/Com) | l | [51,52] |
| | Membrane generation (MeGe) | m | [53] |
| | Nanoparticle enhanced (NP) | n | [54,55] |
| | Electrospray deposition (ESD) | o | [56,57] |
| | Electrical deposition (ED) | p | [58,59] |
| | Ice | q | [60,61] |
| Liquid-to-aerosol conversion (LAC) | | r | [27,62] |
| Hydride generation (HG) | | s | [63,64] |

The letters a–s in the third column are also shown in Figure 3 for ease of comparison.

In order to obtain an overview of the relative frequencies of the different sample preparation methods, all LoDs documented in the literature reviewed were classified according to the method used to achieve them. The results are shown in Figure 4. Most commonly (50%), the sample was first transferred from the liquid to the solid phase to avoid the issues encountered in direct LIBS analysis of aqueous samples (cf. Figure 4a). This was followed by analysis in the liquid phase, with a share of 37%. Conversion to an aerosol with a nebuliser is less common, at 12%. Hydride generation is the least used, but it should be noted that this method is only applicable for certain elements (e.g., As, Sn, Sb, Se, Ge, Pb, Bi, Te) [63,65]. Further classification into subcategories demonstrates that the scientific approaches used by individuals are highly variable (see Figure 4b,c). Liquid bulk (LB) is dominated by the liquid jet (LJ) method (37%), which does not preconcentrate the sample solution but avoids splashing (cf. Figure 4b). The second most common subcategory (30%) is analysis directly on the surface of the liquid, which causes the greatest quantification problems. When analysing inside the liquid (15%), there are slightly fewer physical phenomena that negatively affect the analysis [19]. Soaking up in filter paper can already lead to a slight pre-concentration [37], but this method is less common, at 10%. Analysis of an isolated droplet is probably the most complicated experimental setup within the liquid bulk category and, at 7%, is represented rather rarely. A special form of the liquid jet is the liquid jet capillary, which was used only very rarely with 1%. LSC is dominated by SE (22%) and n-SE (20%) preparation techniques (cf. Figure 4c). Both have in common that they not only convert the aggregate state but also pre-concentrate the sample by evaporation. Therefore, higher sensitivity and lower LoDs are possible with these methods. Ad/ab is slightly less common (18%), which also pre-concentrates the sample, followed by FP (9%). Chelating/complexation (Che/Com), electrospray deposition (ESD), and ion exchange (IE) all have a share of 6%. Electrical deposition (ED) accounts for 5%, and the remaining methods, such as transfer to ice, account for 8%.

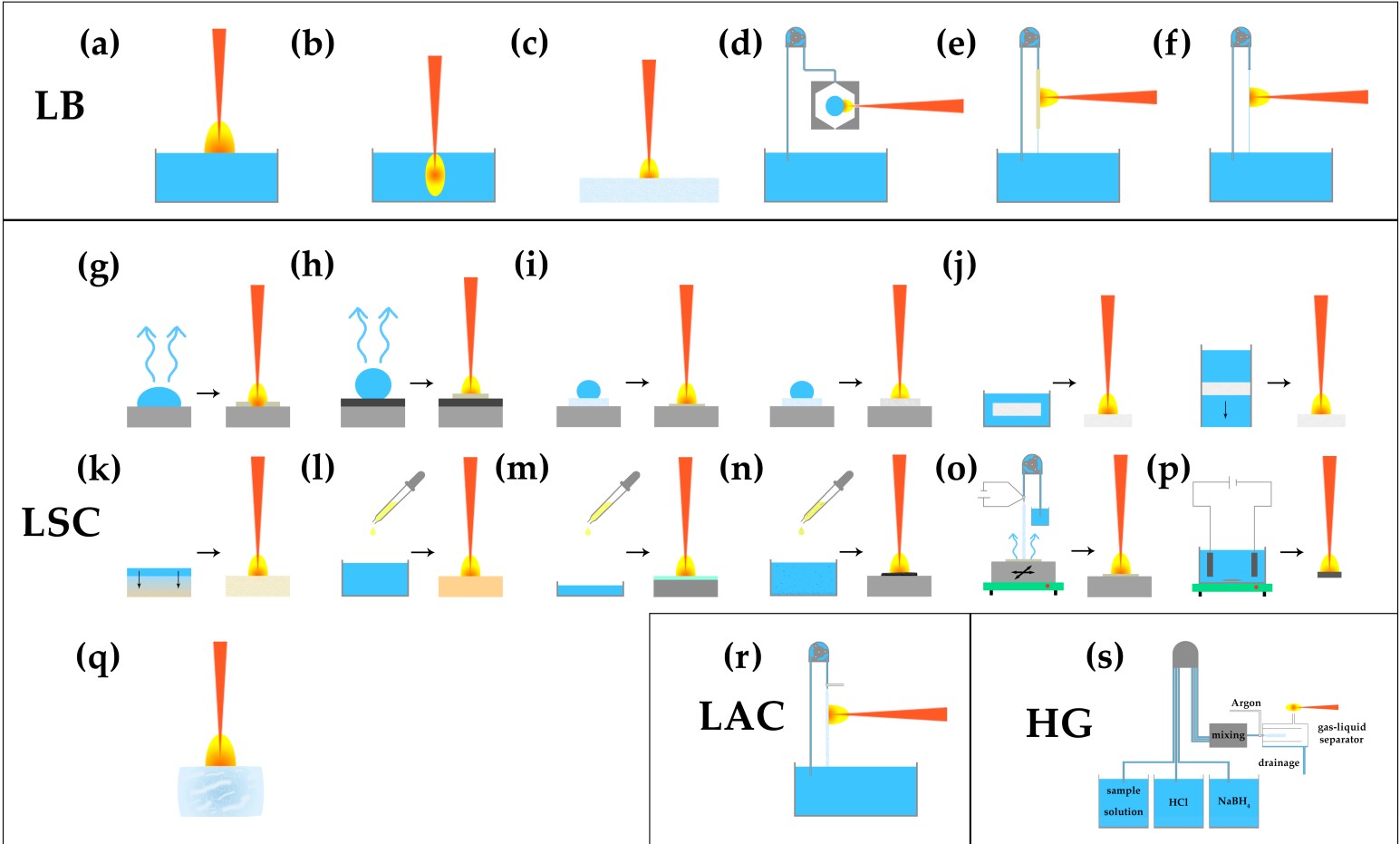

**Figure 3.** Illustration of the most typical sample preparation techniques used in LIBS analysis of aqueous solutions. First row: Liquid bulk (LB): (**a**) surface; (**b**) inside; (**c**) soaked on filter paper; (**d**) isolated droplet (ID); (**e**) liquid jet (LJ) capillary; (**f**) LJ. Second to fourth rows: Liquid-to-solid conversion (LSC): (**g**) non-surface-enhanced (n-SE) evaporation residue (n-SE EvR); (**h**) surface-enhanced (SE) EvR; (**i**) filter-paper-supported EvR—left: analysing the EvR, right: analysing the dried filter paper; (**j**) ion exchange (IE)—left: immersion in solution, right: flowing through a medium, (**k**) adsorption/absorption (Ad/Ab); (**l**) chelating/complexation (Che/Com); (**m**) membrane generation (MeGe); (**n**) nanoparticle-enhanced (NP); (**o**) electrospray deposition (ESD); (**p**) electrical deposition (ED); (**q**) ice;. Fourth row: (**r**) liquid-to-aerosol conversion (LAC); (**s**) hydride generation (HG).

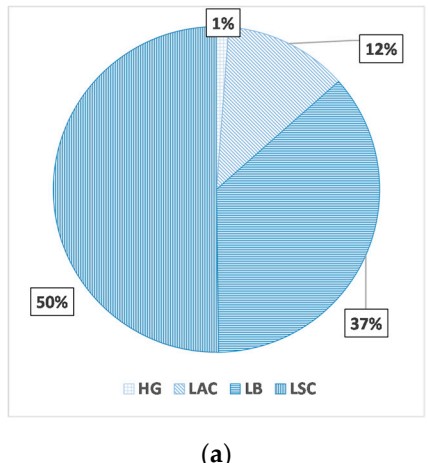
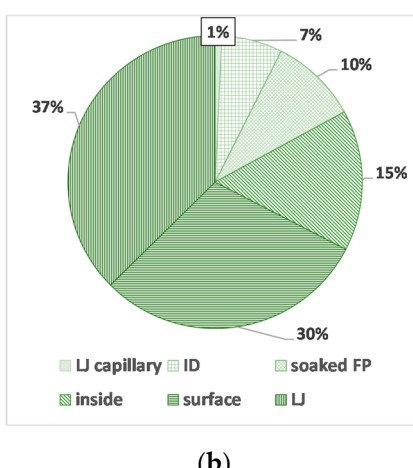
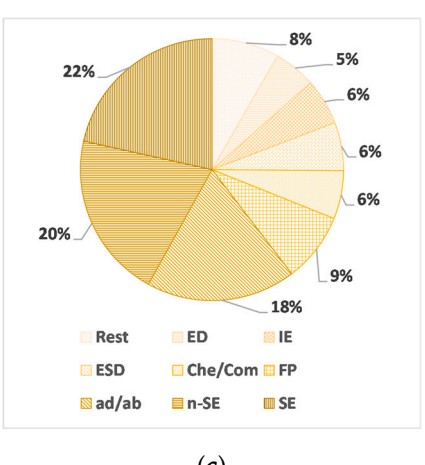

(**a**)  (**b**)  (**c**)

**Figure 4.** Percentages of methods used: (**a**) Percentages for the main categories. HG = hydride generation, LAC = liquid-to-aerosol conversion, LB = liquid bulk, LSC = liquid-to-solid conversion. (**b**,**c**) Percentages for subcategories for the two most common methods used: (**b**) LB, (**c**) LSC. LJ = liquid jet, ID = isolated droplet, FP = filter paper, LJ = liquid jet, ED = electrical deposition, IE = ion exchange, ESD = electrospray deposition, Che/Com = chelating/complexation, ad/ab = adsorption/absorption, n-SE = non-surface-enhanced, SE = surface-enhanced. Data based on the literature reviewed.

### 3.4. Instrument Types, Experimental Setups, and Acquisition Settings

The variety of different lasers, laser manufacturers, and LIBS manufacturers as well as their diverse applications have led to the fact that there is no uniformity in any of the possible settings, nor in the experimental setup. As a result, there may be confusion about which types of LIBS devices, experimental setups, and acquisition settings exist. The following three tables provide a brief overview of some of the different instrument types, experimental setups, and acquisition settings (cf. Tables 3–5). Both Tables 3 and 4 provide a short description as well as information on the main advantages and disadvantages of the instrument type or the LIBS setup. For further details, please refer to the listed literature. Table 5 provides an overview of the acquisition settings that influence LIBS analysis the most. The brief descriptions are intended to give a sense of the parameters that influence the analyses. More comprehensive information is found in the listed literature.

**Table 3.** Overview of different LIBS instruments used for the elemental analysis of aqueous solutions.

| Instrument Type | Description | Advantages | Disadvantages | References |
|---|---|---|---|---|
| Laboratory-based | Laboratory analysis | Sensitive, multi-element detection | Usually inflexible, trained staff needed, expensive | e.g., [66] |
| Online | Continuous online analysis | On-site, continuous, real-time, multi-element detection | Needs own power supply, large | e.g., [67] |
| Telescopic | Remote analysis by a telescopic system | Safe, remote, on-site, multi-element detection | Not useful/necessary for every application | e.g., [32] |
| Portable/handheld | In situ analysis possible | Low-cost, on-site, real-time, multi-element detection | Only low pulse energies, less sensitive | e.g., [44] |

**Table 4.** Overview of different experimental setups for signal enhancement in the elemental analysis of aqueous solutions.

| LIBS Setup | Description | Advantages | Disadvantages | References |
|---|---|---|---|---|
| Double-pulse (DP) instead of single-pulse (SP) | First pulse ablates and generates plasma; second pulse reheats the plume | Higher sensitivity, fast and easy sample preparation, remote and in situ utilisation possible, no tuning of the laser wavelength, simultaneous multi-element detection | Not as sensitive as LIBS-LIF or RE-LIBS | [68,69] |
| Laser-induced fluorescence (LIF) | First pulse ablates and generates plasma; second pulse is tuned to specific analytes | Higher sensitivity due to resonant excitation and background-free signal detection, no spectral interference | No simultaneous multi-element detection, tuneable laser and experienced staff needed | [70,71] |
| Resonance-enhanced (RE) | First pulse ablates and generates plasma; second pulse is resonant with the major species line | Higher sensitivity, simultaneous multi-element detection, low sample consumption | Tuneable laser and experienced staff needed | [69] |
| Resonant (R) | One laser source tuned to specific resonant transition | Simpler experimental setup compared to LIF and RE, higher sensitivity compared to SP, simultaneous multi-element detection | Tuneable laser and experienced staff needed | [69,72] |
| Microwave enhancement (MW) | Enhancement by extended plasma lifetime through mobilised free electrons and ions | Higher sensitivity for a specific element | Complicated setup, requires a microwave system, no simultaneous multi-element detection | [28] |

**Table 5.** Overview of the acquisition settings that most influence the analysis in LIBS analysis.

| Acquisition Settings | Unit | Short Description | Reference |
|---|---|---|---|
| Repetition rate | Hz | An increased repetition rate allows faster analysis and greater averaging for a better signal-to-noise ratio and influences the energy delivered | [13] |
| Pulse energy | mJ | Higher energy results in more and faster ablation | [73] |
| Pulse duration | ns | ns/ps/fs pulses possible; influences the results due to effects such as plasma shielding | [74] |
| Gate delay | ns | Gating can improve the results due to less continuum radiation and therefore better S/N with longer delays | [13] |
| Atmosphere | - | Influences results: Ar > air > He in terms of intensity, plasma temperature, and electron density; He is better than S/N | [75] |
| Wavelength | nm | More energy can be delivered at shorter wavelengths to break bonds and ionise | [73] |

The most important acquisition settings of the laser are probably the wavelength, the pulse energy, the irradiation intensity (focused pulse power density), and the spatial beam quality [13]. The latter can hardly be compared based on the presented data in the publications. The irradiation intensity is determined by the pulse duration in combination with the pulse energy and the repetition rate. As it is only rarely stated, no comparison was made for the reviewed literature. The wavelengths and pulse energies used are compared below. In Figure 5, the frequencies of different wavelengths used in the literature on aqueous analysis by LIBS is shown. It is clear that the most commonly used laser wavelength is the fundamental 1064 nm one for a Nd:YAG laser, followed by the second harmonic 532 nm. Other wavelengths like 266, 352, 500, and 800 nm are less typical, and no reference used the fifth harmonic. Few references were found using lasers other than Nd:YAG. For example, a Ti:Sapphire laser used an 800 nm wavelength for femtosecond analysis [35,76], whereas a pumped dye laser used 500 nm [29].

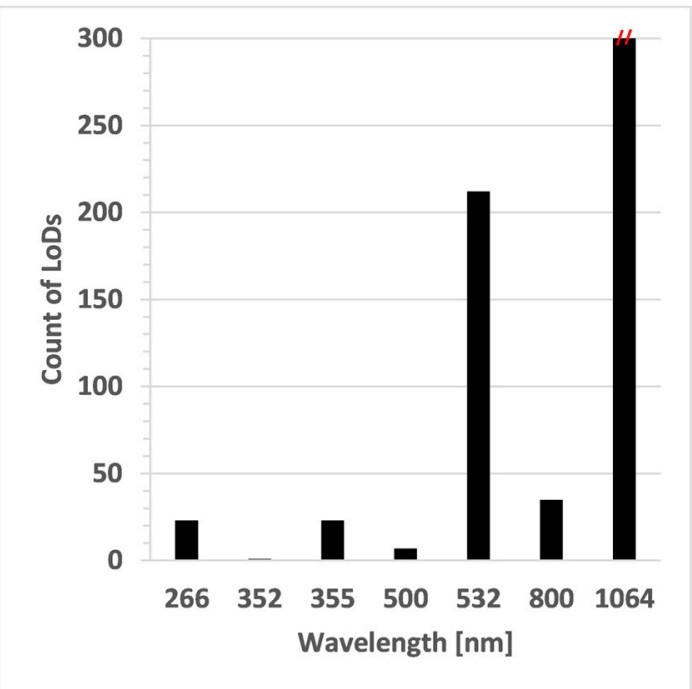

**Figure 5.** Frequencies of different wavelengths used in the literature on aqueous analysis by LIBS. The wavelength is counted more than once per citation if different settings were used, resulting in different LoDs. The data for 1064 nm have been partially hidden (red lines) for the sake of clarity (N = 435). Data based on the literature reviewed.

The use of pulse energy depends on the instrument type, as different lasers are used within portable, telescopic, standoff, and online instruments. The pulse energy cannot always be changed directly by the user. However, it can be changed indirectly via the frequency or the wavelength. Due to the different LIBS instruments used, the different applications, and the possibility of adjusting the pulse energy, a wide range of different pulse energies has been applied as documented by the literature. These range from a fraction of a mJ to several hundred mJ. The highest pulse energy found in the literature reviewed was 800 mJ. However, such high pulse energies are less frequently used, and certain clusters are noticeable with significantly lower energies. Figure 6 shows the pulse energies classified in steps of 10 mJ up to 250 mJ. Energies above 250 mJ are rather unusual, and only 12 LoDs were reported for these energies. By comparison, 635 examples could be identified for energies lower than 250 mJ. The pulse energy of 100 mJ was found particularly frequently (74 times), and in the range below 70 mJ many different pulse energies were used. However, the range 0.1–9.9 mJ was used more (99 times) than the range 100–109.9 mJ (74 times). Data based on the literature reviewed.

### 3.5. Calibration Techniques and Spectral Treatment (Chemometrics)

Different calibration techniques are not discussed in detail in this review but can be found in the literature [77–84]. Spectral processing and data handling are, similarly, covered in detail elsewhere [85–87]. Both calibration techniques and spectral treatment have an influence on the LoD achieved, but it is impossible to evaluate the influence as the calibration and spectral treatment methods used cannot be compared between different references on the basis of the information given in the publications.

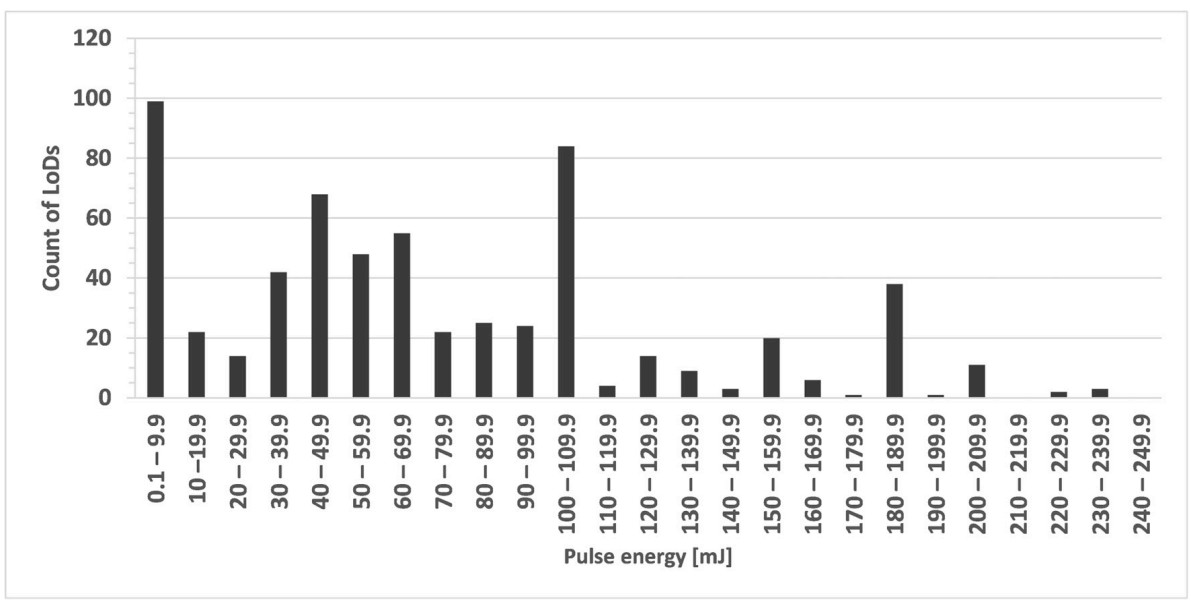

**Figure 6.** Classified frequencies of different pulse energies in the range 0.1–250 mJ (intervals of ten 0.1 mJ steps) used in the literature on aqueous analysis by LIBS. The pulse energy is counted more than once per citation if different settings were used, resulting in different LoDs. Pulse energies above 250 mJ and up to 800 mJ are omitted for clarity.

### 3.6. Self-Absorption and Self-Reversal Correction

Self-absorption takes place when some of the emitted radiation is re-absorbed before it exits the source. As a result, this re-absorbed radiation eventually reaches the detector [86]. In quantitative LIBS analysis, self-absorption is usually manifested by broadened spectral lines in the spectra and by non-linear calibration lines in calibration. Higher concentrations of the analyte no longer result in a proportionally higher intensity [86]. This makes calibration difficult, especially for large concentration ranges. Various methods have been developed to reduce the effect of self-absorption. However, the effects of the different methods cannot be compared based on the reviewed data. Reference should be made here to the extensive literature concerning self-absorption [86,88–92]. Self-reversal can arise independently of self-absorption when there are spatial gradients in plasma temperature and electron density, which typically occur at the edge of the plasma. This effect appears as a confined dip on the top of the emission line [86], which looks like two very close and not individually resolved peaks.

### 3.7. Elements Analysed by LIBS in Aqueous Solutions

Based on the literature reviewed, 56 different elements have been analysed in aqueous solutions using LIBS. Of these, three elements are radioactive (Tc, Th, U), and, thus, at least 53 out of 80 stable elements (66.3%) have been analysed with LIBS in water to date. The most LoDs are reported for Cr (N = 87) and Pb (N = 76), followed by Cu (N = 49) and Cd (N = 43) (cf. Figure 7). For 30 elements, five or more LoDs were identified (cf. Figure 7, marked in dark green). None of the noble gases, nor H, C, O, or Sc, has been analysed to date. Also, some elements of the fifth period (Nb, Rh, Pd, Te, I) and several of the sixth period (Tb, Dy, Ho, Er, Tm, Lu, Hf, Ta, W, Os, Ir, Pt, Tl, (Bi)) have not been analysed.

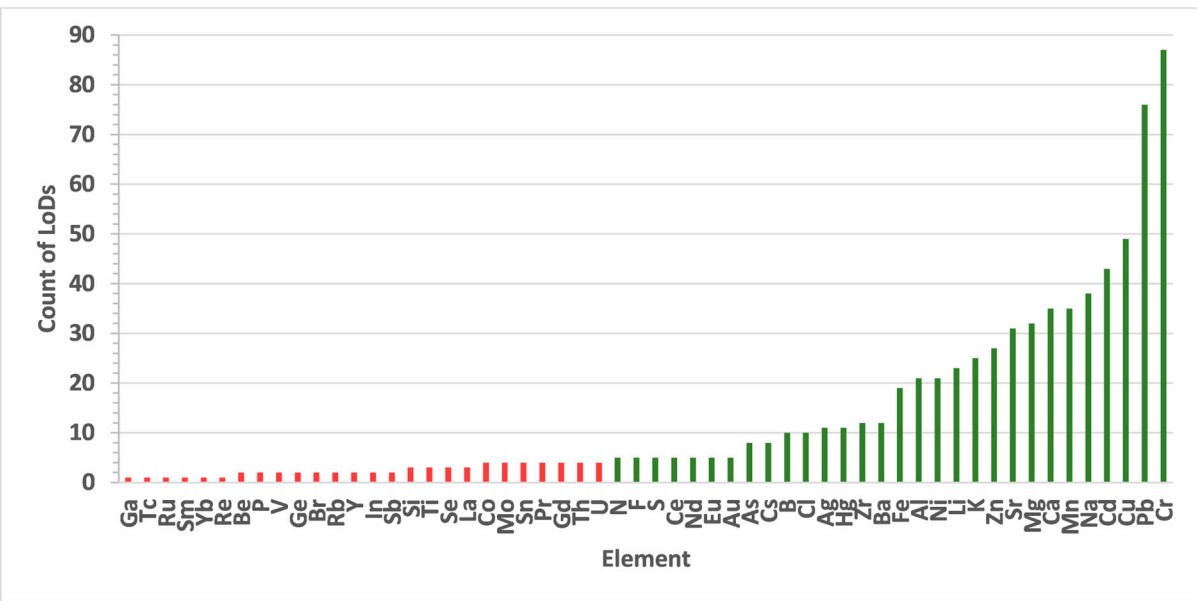

**Figure 7.** Number of detection limits (LoDs) per element in LIBS analysis of aqueous solutions per year. Red: less than five LoDs found; dark green: five or more LoDs found. Data based on the literature reviewed.

*3.8. Detection Limits Achieved*

In Figure 7, the elements are highlighted depending on whether more or less than five LoDs were found and therefore included in the evaluation. The same colour code has been used in Figure 8, which shows the median LoD values for all elements. Dark red bars thus indicate that the median is less reliable than with green bars, as only very few studies down to one detection limit were included in the calculation. It can be seen that by far the lowest LoD median was obtained for Ag. This is followed by V, for which only two detection limits could be used for calculation. Ni, Cu, Li, and Cr also show very low median LoDs with good sample sizes. The worst median LoDs were obtained for F, Tc, U, In, Cl, Yb, and Br, in ascending median order.

(**a**)

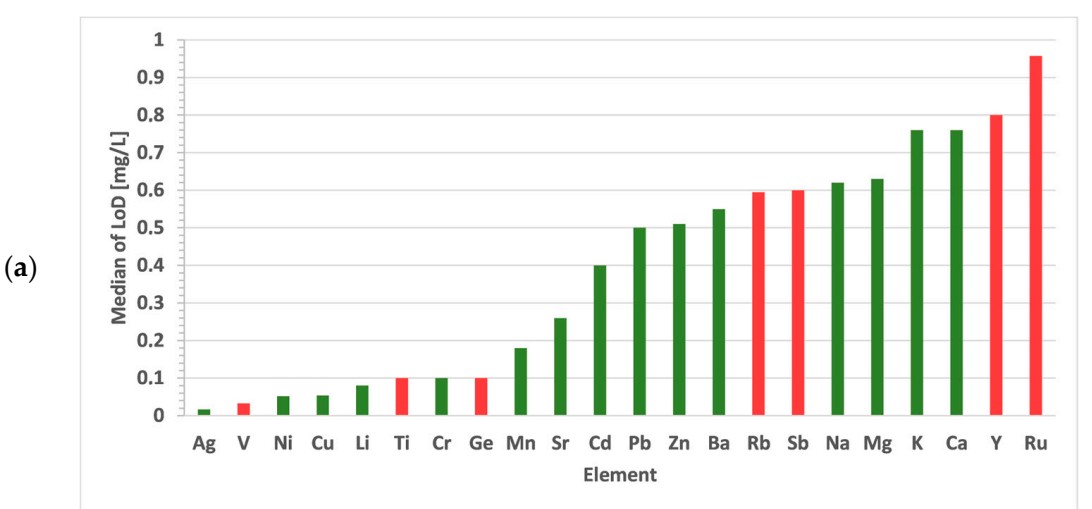

**Figure 8.** *Cont.*

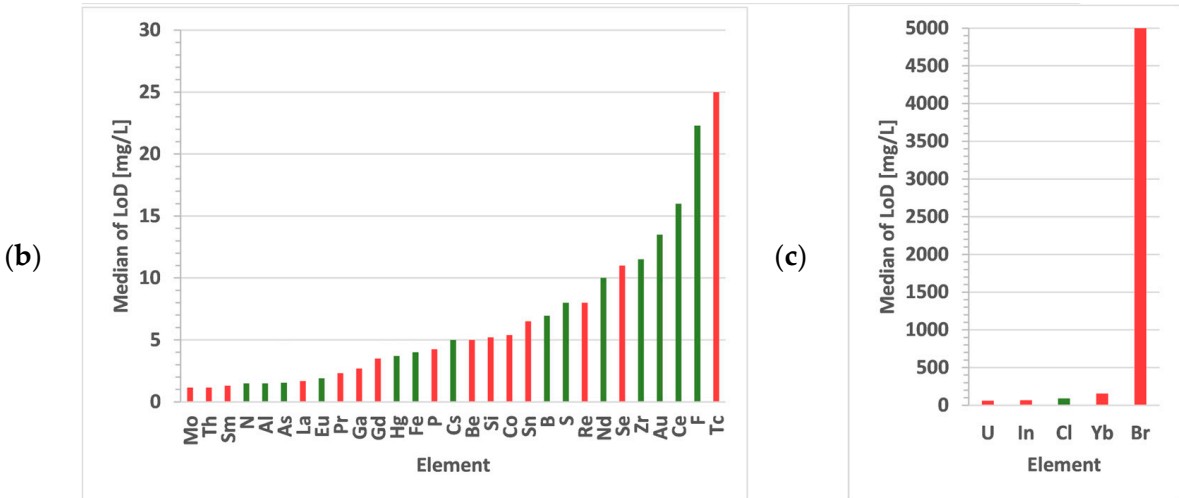

**Figure 8.** Median LoDs per element in LIBS analysis of aqueous solutions. (**a**) Median < 1 mg/L; (**b**) Median < 30 mg/L; (**c**) Median > 30 mg/L. Red: less than five LoDs included for calculation; dark green: more than five LoDs included for calculation. Data based on the literature reviewed.

Figure 9 compares the LoDs obtained with different categories of sample preparation for the two elements with the most comprehensive datasets: Cr and Pb. Here, both elements show significantly lower LoDs for LSC than for LAC or LB for most of the reported LoDs. As HG was only used for Pb and only once, it is difficult to classify it among the other methods. For Pb, there may be a clear order, with LAC giving the highest LoD, LB intermediate, and LSC the lowest. However, there are only five LoDs for LAC, and for Cr LB is worse than LAC.

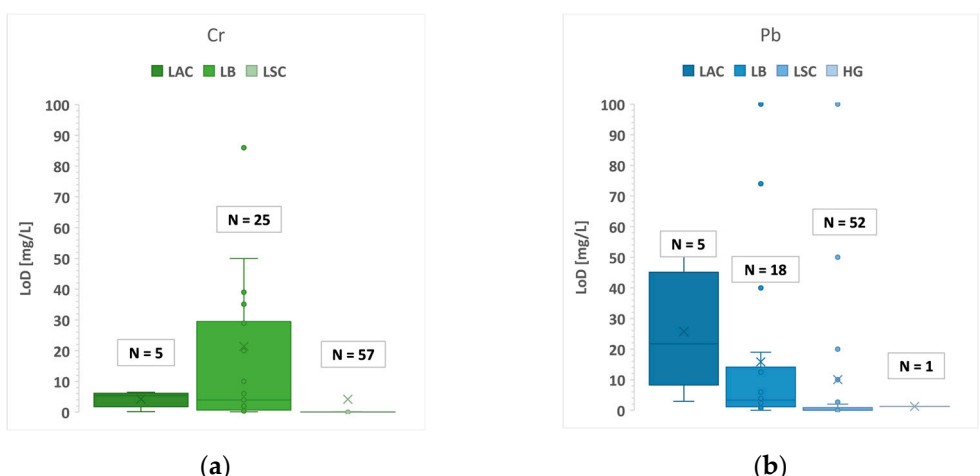

**Figure 9.** Comparison of detection limits (LoDs) achieved in the reviewed literature between different categories of sample preparation: (**a**) Cr, (**b**) Pb. Two values have been faded out for Cr (LB 200 mg/L and LSC 230 mg/L) and one for Pb (LSC 200 mg/L) for the sake of clarity. Data based on the literature reviewed.

Figures 10–13 show the range of LoDs using boxplots for the 56 different elements for which detection limits have been found (excluding the transition metals). In general, both the first- and the second-group elements show medium to high analytical performance according to the classification scheme. In Figure 10, some trends are visible: Li shows by far the lowest LoDs within the alkali and alkaline earth elements, with outliers below 2 mg/L, and therefore it is the most sensitive element within the two groups. For Na, the analysis is less sensitive, and K seems to be somehow problematic to analyse. Cs and Be have the highest LoDs within the first and second main groups, but also, together with

Rb, the lowest data quality (N = 2–8). With the exception of Be, the detection limits for the alkaline earth elements are relatively similar to each other, while the analysis for Sr is somewhat more sensitive.

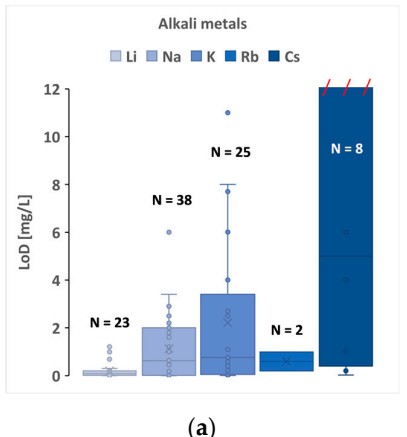 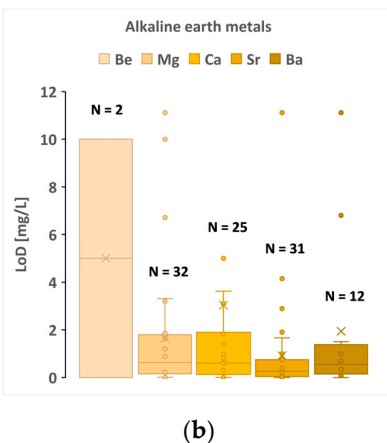

**Figure 10.** Ranges of LoDs for (**a**) the alkali metals and (**b**) alkaline earth metals. Cs has been partially dismissed for the sake of clarity. Data based on the literature reviewed.

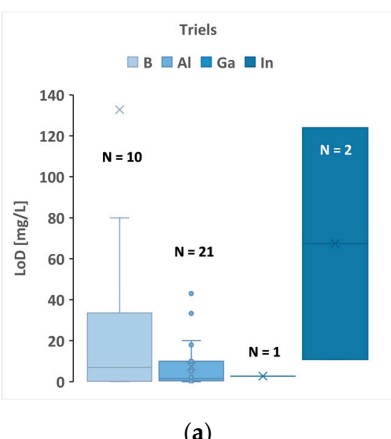 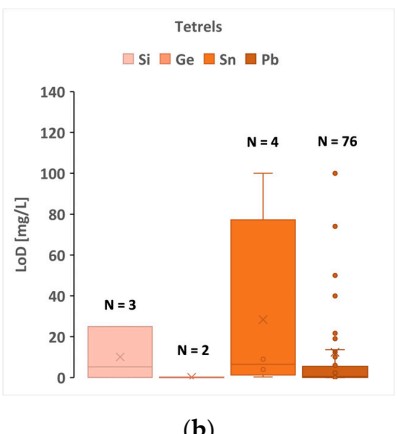

**Figure 11.** Ranges of LoDs for (**a**) triels and (**b**) tetrels. In has been partially hidden for the sake of clarity. Also, one LoD of B and one of Pb have been omitted (1200 and 200 mg/L). Data based on the literature reviewed.

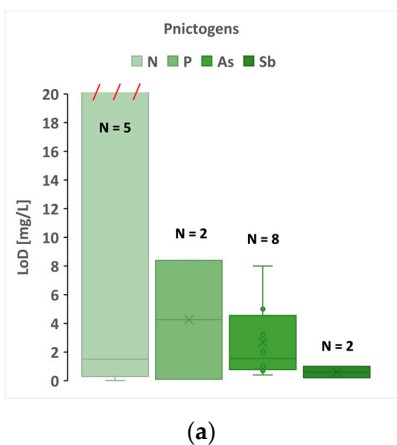 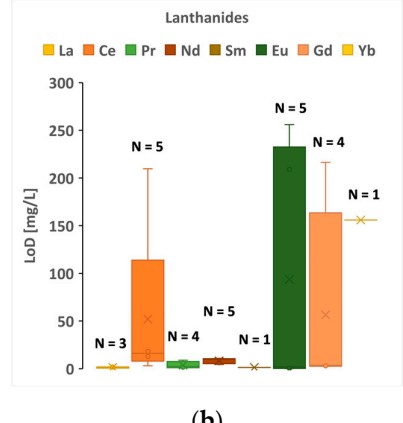 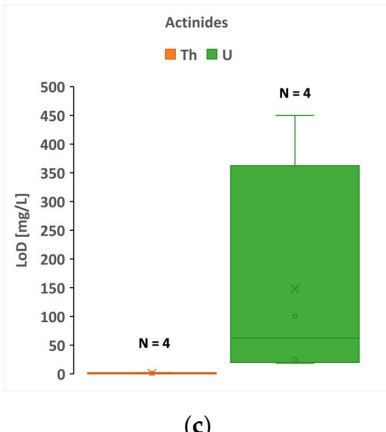

**Figure 12.** Ranges of LoDs for (**a**) pnictogens, (**b**) lanthanides, and (**c**) actinides. N has been partially hidden for the sake of clarity (107 and 542 mg/L). Data based on the literature reviewed.

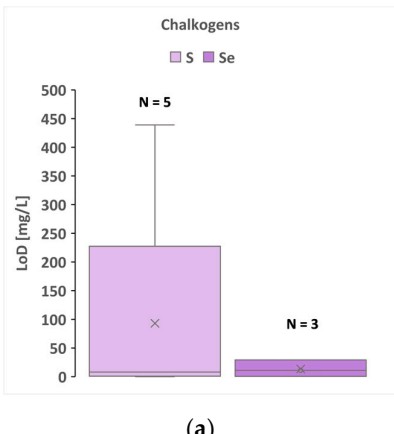
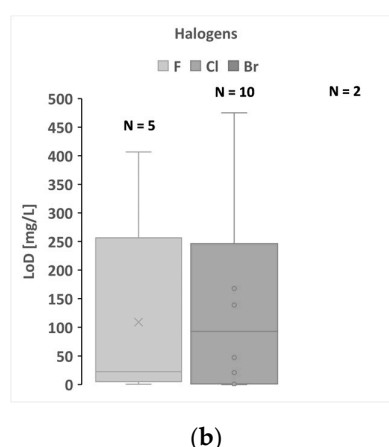
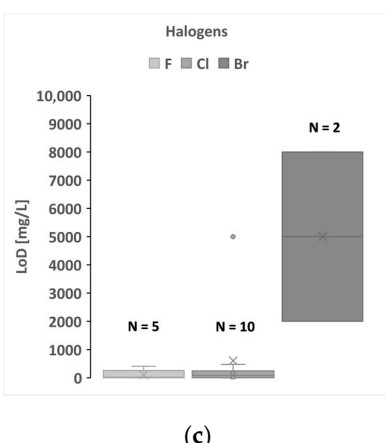

(**a**)  (**b**)  (**c**)

**Figure 13.** Ranges of LoDs for (**a**) chalkogens and (**b**,**c**) halogens. Data based on the literature reviewed.

For the third- and fourth-group elements, LIBS analysis in aqueous solutions generally appears to be less sensitive (cf. Figure 11). Low to high analytical performance can be observed for elements from Group 3 and Group 4. Ga, In, Si, Ge, and Sn are rarely analysed, and less than five LoDs have been found. It is therefore not possible to make any reliable statements other than a trend. B and Al are more frequently analysed. For Al, indifferent analytical performance is observed, with outliers up to 43 mg/L. B shows an outlier of 1200 mg/L, but the same reference also achieved an LoD of 80 by using two laser pulses instead of a single one [19]. Pb may be an exception in both groups, but it was also analysed particularly frequently (N = 76) and the LoDs are very scattered. The highest LoD was 200 mg/L; however, most LoDs were below 15 mg/L and the median LoD was 0.5 mg/L. Therefore, high analytical performance can be reported for the analysis of Pb.

In general, elements of Group 4 (pnictogens) seem to be analysed at greater sensitivity than those from Group 3 (Triels) or Group 4 (Tetrels) (cf. Figure 12a). However, they are analysed rarely, and no reliable statement can be made for P or Sb. N was analysed only five times, with two very high LoDs (107 and 542 mg/L) but also with three LoDs below 2 mg/L. For As, the highest LoD was 8 mg/L and the median was 1.55 mg/L. In lanthanide analysis, very different analytical performance can be observed (cf. Figure 12b). La, Pr, Nd, and Sm were generally analysed with low LoDs and Ce, Eu, Gd, and Yb with high LoDs. However, only a few references are available for all lanthanides. For La, Pr, Sm, Gd, and Yb, less than five LoDs were found. Unsurprisingly, within the actinides, only Th and U were analysed (cf. Figure 12c). Unfortunately, less than five LoDs were found for both, and all LoDs for Th were found in one reference. However, this reference also analysed U, resulting in significantly higher LoDs (18.5 and 24.6 mg/L) than for Th (0.0007–2.25 mg/L). This suggests that Th is probably easier to detect than U.

Both the chalcogens and the halogens, in general, show poor analytical performance (cf. Figure 13). Some of the detection limits were achieved indirectly via other elements or compounds. Although relatively few detection limits were found for the elements within the two groups, it can be assumed that these reflect a trend that only poor analytical performance can be achieved for these elements. The transition metals, as a group of elements, show no clear trend, and are therefore not shown here. However, the fourth period, with elements such as Cr, Ni, Cu, and Zn, shows comparatively low LoDs.

Figure 14 summarises the collected data in the form of a periodic table of elements. The median of the detection limit is given for each element, if one was found. The coloured highlighting, with green for median detection limits up to and including 1 mg/L, light green up to and including 5 mg/L, and red above 5 mg/L, quickly shows which areas of the periodic table can be analysed well with LIBS and which are rather poor.

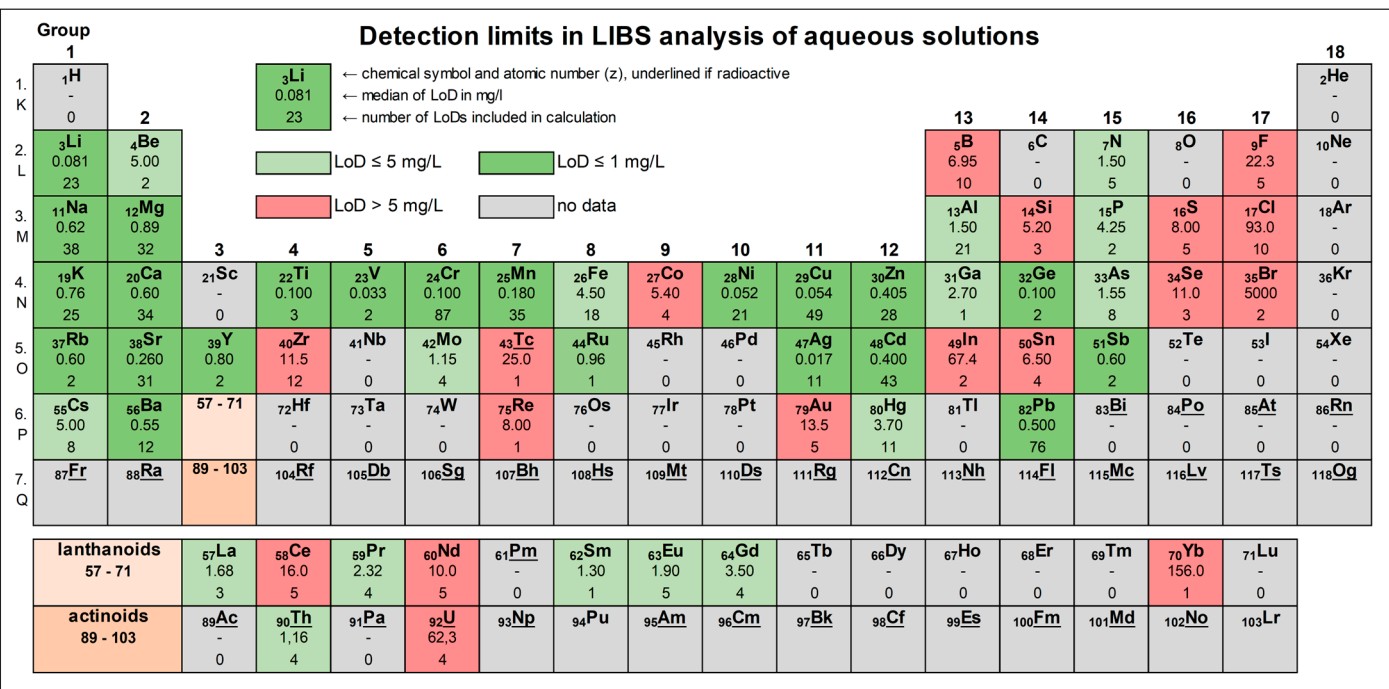

**Figure 14.** Median detection limits reported in a periodic table of elements. The median value of the reported LoDs is plotted beneath the element. As the same numbers of detection limits were not found for each element, the number of detection limits included in the calculation is also given to indicate the data quality. Threshold values have been set for faster assessment. Median detection limits below 1 mg/L are marked green, up to 5 mg/L light green, and above 5 mg/L red. The underlying periodic table was created freely according to [25].

## 4. Discussion

### 4.1. Sample Preparation Techniques

It has been shown that there is no standard approach to aqueous LIBS analysis in terms of sample preparation, LIBS experimental setup, or acquisition settings (cf. Table 2, Figures 3–6). Several very different techniques are used, each with its own advantages and disadvantages (cf. Tables 2–4). This is due to the variety of possible applications, ranging from basic research in laboratory analysis, typically using standoff or self-arranged LIBS [66], to remote analysis for hazardous substances, using telescopic systems [32], and to in situ analysis for environmental concerns using online [67] or portable instruments [11]. Therefore, both the LIBS setup and the sample preparation technique should be selected according to the desired application.

However, one of the sample preparation methods listed in Table 2 can be excluded in the choice of sample preparation. Surface analysis of bulk liquids has many limitations, such as high laser energy requirements and low sensitivity, which have been extensively described in the literature (e.g., [93]), and is therefore not recommended. Splashing and liquid evaporation compensation methods should be used instead. Analysis inside the liquid is a bit more promising but is subject to shot-to-shot variability [36].

Furthermore, in situ applications require sample preparation techniques that are easy to use in the field. Therefore, methods such as LB soaked on FP, SE LSC, and LSC FP are recommended. However, when selecting a sample preparation technique, it should be noted that several sample preparations also pre-concentrate the sample solution and thus can lower the LoDs significantly (e.g., most LSC techniques). This can help to achieve LoDs significantly below 1 mg/L. As portable instruments are typically less sensitive than laboratory devices using higher energies, sample preparation should include preconcentration steps. For online or laboratory analysis, more complex LIBS setups with more complicated sample preparation techniques can be chosen (e.g., LJ, LAC). Here, preconcentration may

be helpful but not imperative. Some LJ analysis already reached sensitivities comparable to commercial ICP-AES devices without any preconcentration [7] and could also be used in online and real-time analysis. For Na, cylindrical jets showed lower sensitivity compared to sheet jets [7]. The special form of an LJ capillary effectively reduces splashing [40] but brings along a more complicated experimental setup and costly consumables.

When only hydride-generating elements (e.g., As or Se) are to be analysed, HG is a good choice as it significantly improves sensitivity by eliminating spectral and chemical interference [63]. However, this method is limited to a small group of elements, which can form hydrides (e.g., As, Sn, Sb, Se, Ge, Pb, Bi, Te), and it requires a complicated measurement setup.

Coupling LIBS with ion exchange techniques can also be a good option to improve sensitivity and reduce matrix effects in LIBS liquid analysis [47]. However, this also requires a complicated setup, which reduces the advantage of simple and fast sample preparation over other methods such as IC. The same is true for the isolated droplet technique. An easy way to avoid splashing is the LSC technique of freezing the sample [60,61]. This sample preparation does not require much expertise or a complicated setup and could also be performed in situ using liquid nitrogen [61]. However, no preconcentration is achieved this way.

ESD achieves preconcentration, but many chemicals are required, as well as additional parts and experimental knowledge. ED also pre-concentrates the samples and therefore has improved sensitivity [58,59]. The main drawback is the comparatively long time required for analysis due to the deposition process (>>10 min). Ad/ab on wood chips [49] or graphene oxide [50] is also quite time consuming but can be realised cost-effectively and improves sensitivity as well. Chelating (e.g., [51]) is even more time-consuming, requires more chemicals, and is usually not possible for multi-element analysis. However, it can be more efficient in enrichment than FP [51]. MeGe, as described by [53], is very similar to chelating, but requires even more time in sample preparation due to the long drying process. This makes the method less favourable in comparison to the other presented methods.

Analysis in ID, either with an isolated droplet generator (IDG) or by acoustic levitation [38], adds complexity in the experimental setup, which minimises its advantages over conventional analytical techniques. NPs have the potential to improve the sensitivity as well [54,55] and are easier to introduce in sample preparation.

When using LSC of a droplet on a surface with no adsorption/absorption step, some form of SE is recommended to improve the homogeneous distribution and shape of the evaporation residue (EvR). Without any SE, an effect typically referred to as the coffee ring effect (CRE) occurs, leading to lowered repeatability due to inhomogeneous distribution of the EvR [44,72]. SE can be implemented by making the surface more hydrophobic to improve the distribution of the EvR [44] to introduce a geometric constraint (GC) technique [94,95] or to combine both [55].

### 4.2. Instrument Types, Experimental Setups, and Acquisition Settings

Furthermore, it is also possible to improve the various sample preparations with a special LIBS setup. These include, e.g., double-pulse (DP), laser-induced fluorescence (LIF), resonance-enhanced (RE), resonant (R), and microwave enhancement (MW) LIBS. The main advantages and disadvantages of the special setups are presented in Table 4. Depending on the design of the LIBS, such setups may not be possible or lead to complexity that diminishes the main advantage of LIBS over other laboratory methods in being simple in sample preparation [96]. The simplest method to use is DP, which typically enhances the signal ten times and for some elements up to 50 times [96]. The other experimental setups are more complicated, require more experienced staff, and do not all allow multi-elemental detection, but can improve sensitivity (cf. Table 4).

The acquisition settings (cf. Table 5) also affect sensitivity and should likewise be adapted to the method. For example, the gate delay should be long enough to minimise

the influence of continuum radiation, which typically occurs at the very beginning of the breakdown and provides no useful information, in addition to degrading the signal-to-noise ratio [26]. However, with overly long gate delays, the signal becomes weaker. Although it is possible to set the optimum delay time for each element individually, using different delay times for different elements prevents simultaneous detection. For this reason, when analysing multiple elements, a gate delay should be selected that is suitable for all the elements to be detected.

Another enhancement possibility, which is often easy to implement, is to use a purge gas such as Ar, which also amplifies the signal intensity significantly [75]. Even some portable LIBS units are already equipped with Ar cartridges for signal amplification [44].

A change in pulse duration, pulse energy, wavelength, or repetition rate essentially influences the energy that is supplied to the sample and is thus available for the formation of the plasma (cf. Table 5). Moreover, increasing the repetition rate reduces analysis time and improves averaging [13]. Typical lasers used in LIBS liquid analysis are Nd:YAG lasers with a fundamental wavelength of 1064 nm (cf. Figure 5). Using different wavelengths can improve detection. For example, the $CO_2$ wavelength of 10,600 nm is particularly suitable for the analysis of water, as it absorbs in the infrared range [13]. However, the wavelengths used were between 266 and 1064 nm (cf. Figure 5). The wavelengths 532, 355, (352), and 266 nm are the second to fourth harmonics of the Nd:YAG laser, with a fundamental wavelength at 1064 nm.

According to [13], typical pulse energies for LIBS are between 10 and 500 mJ. In the literature reviewed, pulse energies between 0.2 and 800 mJ were found (cf. Figure 6), with most in the range 0.2 to 100 mJ. Energies below 10 and around 100 mJ are the most common (cf. Figure 6). The pulse energy must be high enough to ablate enough material to produce a strong signal. However, a low energy pulse combined with a shorter pulse width and higher repetition rate will also result in a higher power density [13]. Handheld devices generally operate with lower pulse energies. For example, the SciAps Z300 uses a pulse energy of 5–6 mJ, a frequency of 50 Hz (10 Hz), and a pulse width of 1 ns [97].

### 4.3. Elements and Their Reported Detection Limits

Looking at the elements analysed, several areas of interest stand out (cf. Figure 7). The frequent testing of potentially toxic elements [98] such as Cr (N = 87) or Pb (N = 76) is particularly striking. There is a very strong interest in the detection of these elements because they are problematic elements in drinking water [99]. Also, Cu and Cd are analysed very frequently, which are also potentially toxic elements [99]. Furthermore, there is a second group of elements that are often analysed: Na, (Mn), Ca, Mg, Sr, K, and Li. These are typical cations in natural waters and therefore of interest in natural water analysis.

It is striking that arsenic (As) is less often analysed then Cr, Pb, Cu, Cd, and Zn, although it is a very problematic element in drinking water [100]. This may be because As is not as well analysed by LIBS as other elements such as Li and Cr. According to Cremers and Radziemski, the most common and useful spectral range for LIBS analysis is 200–900 nm [13]. Unfortunately, there are few strong lines for As in this range that can be used for quantification. Therefore, for As quantification, it is recommended to use pre-concentration methods or to choose HG.

S, N, F, Br, and P (and to a lesser extent Cl) are also relatively rarely analysed, although they are typically present as anions in natural water. This is also due to the fact that these elements are not analysed as well by LIBS as other elements such as Li and Cr. These six elements have among the highest ionisation energies of the 56 elements for which detection limits have been found. The ionisation energy plays a major role for these elements, since the lines typically chosen for quantification lie in or near to the infrared range (except for P) and are difficult to excite due to the high ionisation energy [101]. The intensities are therefore usually so weak that low concentrations cannot be detected [101,102]. For example, the strongest lines for F and Cl are below 200 nm, below the range of typical CCD detectors and in a range where a vacuum is required in the light path [103]. Therefore,

a detour via indirect analysis is usually used (for example, molecular emission, as compounds such as CaF and CaCl show higher intensities) [14]. Another option is to analyse the excess of a more easily analysable element (e.g., Ag or Ba) that reacts with the anions by precipitation [101].

It is also interesting to note that some atypical elements such as Tc were analysed in water. The reason for this is the possibility of remote LIBS analysis. Samek et al. analysed Tc with a telescopic LIBS system over a distance of 3 m [32]. Also, the catastrophe of the Fukushima nuclear disaster has led to development of LIBS aqueous analysis for elements like Cs, Sr, and Zr [43,104–106].

### 4.4. Limitations of the Data Review

There are a number of limitations that need to be considered when interpreting the data reviewed, in order to avoid drawing premature conclusions and to be able to interpret the graphs and data, especially the periodic table of elements, correctly. First of all, the review was limited to publications available between 1984 [19] and August 2023. Moreover, although most of the reported LoDs have been calculated using the $3\sigma$-IUPAC criterion, some have been calculated using other formulae, and some may be overly optimistic as they are often incorrectly calculated [84]. In addition, very different instrument types, experimental setups, and sample preparation methods were used in the reviewed literature to achieve the LoDs, including steps of pre-concentration and indirect analysis (e.g., [14,101,107]).

Another factor that should not be neglected is the lasers and spectrometers used, which strongly influence the possibilities of analysis, for example, through different possible energies and resolutions. A distinction should be made between laboratory-based, online, telescopic, and portable devices (see Table 3), if one wants to directly compare the detection limits of two analyses of different LIBS types. Here, too, the application plays a major role. For in situ measurements, higher LoDs are usually accepted if direct decisions can be made on the basis of the measured concentrations and if more detailed laboratory analyses are still possible afterwards [11]. Acquisition settings can either be a preset by the instrument or they can be set by the user. Different gate delays, gate widths, pulse energies, repetition rates, and wavelengths also result in different detection limits.

Furthermore, the atmosphere used in LIBS analysis also plays an important role in the detection limit achieved [75]. For example, Ar and He improve the signal intensity and thus lead to better detection limits. However, various atmospheres were used in the compared literature, leading to problems with comparability.

The use of different calibration techniques affects the calculated LoD, as does the use of different spectral lines (atomic/ionic/molecular) for the same element. This was, for example, clearly shown for Zr in [105]. In addition, it is also possible to use several lines per element and also ratios of different lines for calibration [22] to reduce the negative matrix effects.

When analysing aqueous solutions with several dissolved elements, it is also important to ensure that the lines used for quantification are not excessively close together, as they will be falsely detected as one peak by the sensor (e.g., Zn I 468.01 nm & Cd I 467.81 nm, cf. Figure 15). The choice of the best spectral lines or the best ratios can lead to improved LoDs for the intended application. However, the multiplicity of the lines or of the ratios of the lines used limits the comparability of the LoDs in the literature to some extent.

The difference in the liquids analysed also has an influence on which detection limit is achieved due to their different matrices (cf. [108]). LIBS analysis is very susceptible to matrix effects [22]. Aqueous solutions with complex matrices therefore generally lead to higher detection limits. To avoid this problem, certain liquids were excluded in advance, and only aqueous solutions were considered.

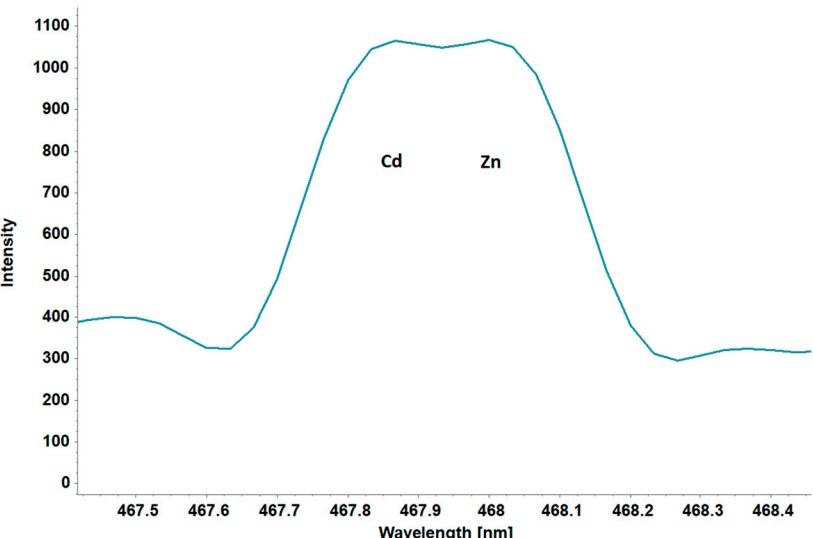

**Figure 15.** The two closely spaced lines of Cd I (467.81 nm) and Zn (468.01 nm) appear almost as a single peak when a solution containing both elements at 125 mg/L is analysed by a portable LIBS using the SE-LSC method described in [11].

Last but not least, a very important factor is the number of references per element. A threshold of five references has been applied to illustrate sufficient data in several graphs (e.g., Figures 7 and 8). However, if these five LoDs per element are reported in one reference for different lines used in the calibration, there is no statistical improvement in comparison to one reported LoD.

Overall, however, the limitations listed cannot be avoided, otherwise the sample size would be reduced further. However, if a sufficient number of LoDs is available for each element, the limitations also become relative. Despite the limitations, this review provides a quick and comprehensive overview and can serve as a guide for further research in this area. Detailed information can be found in the Supplementary Data, which also lists all of the references used.

*4.5. Future Directions*

It is unlikely that LIBS will be widely used in the laboratory for aqueous analysis in the near future, because there are already well-established methods such as IC, AAS, and ICP-MS. However, the possibility of the miniaturisation of LIBS and the handheld or online instruments already available offer a great opportunity. In 2009, there was no truly handheld LIBS instrument [96]. It was only 10 years ago, in 2013, that the first truly handheld LIBS analyser was introduced by SciAps Inc. (Woburn, MA, USA) [16]. Since then, there has been a lot of research with such handheld instruments in the area of solids analysis [97,109–111], but still little in the field of liquids analysis [11,44]. Therefore, there is distinct potential to investigate LIBS as a novel technique in liquids analysis. LIBS may also offer benefits simply through the fact that LIBS analysis allows immediate decision making and pre-selection of samples for laboratory analysis. Also, the miniaturisation of instruments (online and portable/handheld) opens a new field in which LIBS could establish itself, as it is able to simultaneously analyse the entire periodic table of elements in water with comparatively low LoDs. This is a cost-effective and efficient approach compared to investigations solely relying on laboratory analyses. Moreover, pre-concentration methods can significantly improve sensitivity and therefore the acceptance of LIBS.

**5. Conclusions**

This review has provided guidance on how to use LIBS for elemental analysis in aqueous solutions, which elements can be analysed, and which methods achieve low LoDs. Over 153 publications (1984 to August 2023) could be identified that covered the topic

of the elemental analysis of aqueous solutions. Five classes of sample types could be identified, with most research on diluted stock solutions, but also with applications to artificial, biological, natural, and natural saline waters. Much of the research on this topic is currently conducted in China.

Direct liquid analysis using LIBS is prone to low sensitivity due to splashing and cooling of the plasma. Therefore, many sample preparation techniques have been developed or adapted to improve sensitivity. Sample preparation in LIBS aqueous analysis can be classified into four main categories: liquid bulk (LB), liquid-to-solid conversion (LSC), liquid-to-aerosol conversion (LAC), and hydride generation (HG). The most common is LSC, followed by LB and LAC. HG is quite seldom used, but also only applicable to a few elements. The most used subcategories within LSC are surface-enhanced (SE), non-SE, and adsorption/absorption (ad/ab). Within LB, the most used subcategories are liquid jet (LJ), surface, and inside.

Four different instrument types are used for analysis: laboratory-based, online, telescopic, and portable/handheld. In addition, there are different experimental setups in use for signal enhancement, e.g., double-pulse (DP) instead of single-pulse (SP), laser-induced fluorescence (LIF), resonance-enhanced (RE), resonant (R), and microwave (MW) enhancement.

Also, several acquisition settings strongly influence elemental analysis in aqueous solutions: repetition rate, pulse energy, pulse duration, gate delay, atmosphere, and wavelength. Typically the fundamental wavelength of the Nd:YAG laser is used (1064 nm), but the second harmonic (532 nm) is also used quite often. All other wavelengths have been used quite seldom in the reviewed literature. Very different pulse energies have been used in the literature; however, 100 mJ was used the most. The range from 0.1–9.9 mJ was used even more frequently than the range 100–109.9 mJ. In total, 56 different elements have now been analysed in aqueous solutions using LIBS, including three radioactive elements.

Most LoDs have been reported for Cr and Pb, and for 30 elements five or more LoDs are reported. In particular, the analyses of Ag, Ni, Cu, Li, and Cr performed well in the literature reviewed, with median LoDs less than or equal to 0.1 mg/L and more than ten LoDs included in the calculation. The analyses of Mn, Sr, Cd, Pb, Zn, Ba, Na, Mg, K, and Ca also performed quite well in the literature reviewed with median LoDs lower than or equal to 1 mg/L and more than five LoDs included in the calculation. Elements of the 15th–17th group of the periodic table of elements tend to show higher detection limits in the analysis with LIBS, because they typically have higher ionisation energies and, in some cases, only lines in the infrared range can be selected. As a result, the intensities of the selected lines are often not sufficient at low concentrations. This is especially true for S, N, F, Br, P, and Cl, which typically occur as anions in water. It is recommended to choose an indirect determination via molecular emissions or excessive elements by precipitation for these elements.

There is no uniformity in the literature when choosing a sample preparation technique and selecting an experimental setup. This is because there are advantages and disadvantages depending on the choice made, and, therefore, the most appropriate method for the particular application should be chosen. However, the analysis of bulk liquid at the surface gives the worst results and should be excluded. HG only works for hydride-generating elements and is complicated to set up but can lead to a significant improvement in sensitivity. If the setup can be realised, this method can help to quantify hydride-generating elements that are not as well analysed as Cr or Li, such as As.

DP often has an advantage over SP, especially for liquid samples, but not every instrument allows DP. However, the conversion from the liquid phase to a gas or solid phase also has certain advantages. For example, the SE LSC leads to a preconcentration of the sample, as only the EvR, i.e., the elements of interest, are analysed. As a result, the sensitivity increases significantly, and elements with generally poorer sensitivity can be analysed better.

Since the RSD for LIBS analysis is usually quite high, multiple measurements should be considered to increase the accuracy. However, since the analysis time for LIBS is very short and the time can be reduced by increasing the repetition rate, this is not a problem for analytical methods with small sample quantities.

Overall, LIBS is a powerful analytical technique capable of simultaneous multi-element analysis not only of solids but also of aqueous solutions. As long as the limitations are known and certain sample preparations and experimental setups are used, any element can be detected in water. Advantages such as low acquisition cost, rapid and simultaneous detection of multiple elements, and the possibility of in situ or remote analysis are convincing arguments for using LIBS as an alternative to established analytical techniques in the elemental analysis of aqueous solutions.

**Supplementary Materials:** The following supporting information can be downloaded at: https://www.mdpi.com/article/10.3390/spectroscj2010001/s1, Spreadsheet S1: LIBS applied to elemental analysis of aqueous solutions [112–196].

**Author Contributions:** Conceptualization, N.S.; methodology, N.S.; software, N.S.; validation, N.S.; formal analysis, N.S.; investigation, N.S.; resources, B.G.L.; data curation, N.S.; writing—original draft preparation, N.S.; writing—review and editing, B.G.L.; visualization, N.S.; supervision, B.G.L.; project administration, N.S.; funding acquisition, B.G.L. All authors have read and agreed to the published version of the manuscript.

**Funding:** This research received no external funding.

**Institutional Review Board Statement:** Not applicable.

**Informed Consent Statement:** Not applicable.

**Data Availability Statement:** The data presented in this study are available on request from the corresponding author.

**Acknowledgments:** We would like to thank our colleagues and former colleagues for fruitful discussions, especially L.T. Krebbers, K.N. Fiedler, and J.-N. Sander. Moreover, we thank the four anonymous reviewers for their constructive comments, which greatly improved this manuscript.

**Conflicts of Interest:** The authors declare no conflicts of interest.

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
