# Peer review of "Laser-Induced Breakdown Spectroscopy Applied to Elemental Analysis of Aqueous Solutions—A Comprehensive Review"

_2813-446X, doi:10.3390/spectroscj2010001_

Round 1

Reviewer 1 Report

Comments and Suggestions for Authors

This review provides valuable information for future users of Laser-induced breakdown spectroscopy. I think this form is sufficient to publish the Spectroscopy Journal as review after very minor revises as follows.

1. line 39. sent to a [13] is bit strange. I think some words are missing.

2. Count of LoDs is confusing. What is the difference between Count of LoDs and reference shown in Figure 1. If the difference is minor, I recommend the author to use "reference".

3. It is better to explain what is "self-reversal" is.

Author Response

Dear Sir or Madam,

thank you very much for your constructive comments, the implementation of which have improved the manuscript. We have highlighted all changes in blue within the manuscript. Please find our responses to your comments below:

1) line 39. sent to a [13] is bit strange. I think some words are missing.

Thank you for the hint. The word must have disappeared when changing the citation style. We have added “spectrometer” to complete the sentence.

2) Count of LoDs is confusing. What is the difference between Count of LoDs and reference shown in Figure 1. If the difference is minor, I recommend the author to use "reference".

The distinction lies in the fact that a single reference can report multiple LoDs, necessitating the use of different terms. Nonetheless, for the sake of clarity, some modifications have been implemented:

line 110: “one or more” added

line 238 “reporting one or more detection limits for the analysis of aqueous solutions by LIBS from 1984 to date” added

3) It is better to explain what is "self-reversal" is.

Indeed, this term requires an explanation. However, the explanation can be found in lines 419-423: “Self-reversal can arise independently of self-absorption, when there are spatial gradients in plasma temperature and electron density, which can typically occur at the edge of the plasma. This effect appears as a confined dip on the top of the emission line [86], which looks like two very close and not individually resolved peaks.”

Merry Christmas and best regards

Nils Schlatter

Reviewer 2 Report

Comments and Suggestions for Authors

The manuscript entitled "Laser-induced breakdown spectroscopy applied to elemental analysis of aqueous solutions - a comprehensive review" (ID: 2709317) is a review on liquid detection of LIBS. This submission has summarized the current development of LIBS applications for aqueous solution. According to my review, I recommend ACCEPT for this submission, while several corrections are needed.

1) Page 1, Line 39__ there is a missing word in the sentence "and sen to a...";

2) Page 2, Line 60__ missing word after "in 1984 by";

3) Page 11, Fig. 5__ there are 2 red line on the strip of "1064";

4) Page 12, Line 347__ word mistake "atypical";

5) Page 19, Line 569__ CO2 wavelength should be 10.6 microns;

6) Page 20, Line 590__ word mistake "As...then";

7) Please double check the reference. For example, the Ref. 33 and the Ref. 157 are same;

8) If possible, please add the long-pulsed LIBS in this review, because it is also an effective way for aqueous solution (A. Matsumoto, et al. Analytical Chemistry, 2015, 87: 1655-1661) .

Author Response

Dear Sir or Madam,

thank you very much for your constructive comments, the implementation of which have improved the manuscript. We have highlighted all changes in blue within the manuscript. Please find our responses to your comments below:

1) Page 1, Line 39__ there is a missing word in the sentence "and sen to a...";

Thank you for the hint. The word must have disappeared when changing the citation style. We have added “spectrometer” to complete the sentence.

2) Page 2, Line 60__ missing word after "in 1984 by";

Again, the word must have disappeared when changing the citation style. We added “Cremers et al.” in line 62.

3) Page 11, Fig. 5__ there are 2 red line on the strip of "1064";

The red lines should show that data has been partially hidden. For clarity we added “(red lines)” in the description.

4) Page 12, Line 347__ word mistake "atypical";

We have exchanged “atypical” by “unusual”.

5) Page 19, Line 569__ CO2 wavelength should be 10.6 microns;

Indeed, a German decimal separator was the issue. Thank you.

6) Page 20, Line 590__ word mistake "As...then";

There is no mistake, but the element arsenic is being discussed. While using the elemental symbols can lead to misunderstandings, writing out the full name of the element makes the text lengthy and long-winded. In this case, we have written out arsenic for better understanding.

 7) Please double check the reference. For example, the Ref. 33 and the Ref. 157 are same;

Reference number 157 has been removed and the remaining numbers have been rechecked.

8) If possible, please add the long-pulsed LIBS in this review, because it is also an effective way for aqueous solution (A. Matsumoto, et al. Analytical Chemistry, 2015, 87: 1655-1661).

Unfortunately, we did not find this paper at an earlier stage. As adding any further references reporting a LoD would affect the entire database, and thus Figures 1, 2, 4 – 14 as well as many passages in the text, we cannot include it anymore.

Merry Christmas and best regards

Nils Schlatter

Reviewer 3 Report

Comments and Suggestions for Authors

The authors have done great work on the manuscript for its publication in the journal, however, there is a need to clarify the following for its improvement:

Abstract: Simplify the abstract by explaining briefly – introduction (2 – 3 lines), aims/objective (1- 2 lines), methods adopted (2-3 lines), trends/outcomes (3 – 4 lines), conclusion (2-3 lines), recommendation/suggestions for further studies (1 – 2lines)

Line 15: Mentions some examples of the LIBS instrument types

Line 17: which methods are documented?

Line 20 – 21: Unclear statement

Line 22: Include the full meaning of the keyword be an abbreviation

Line 102 – 103: Sources of the data for the figure should be clearly stated

Line 104: include some statement about methodology in the abstract

Line 124 – 123: The focus here is different from the objective stated in the abstract (Line 13 - 15). Your work should be consistent and comprehend with each other

Line 130 – 131: contracting statement. Kindly clarify

Line 133: Include publication year range

Fig. 2: Source and year? Can you explain the necessity of this figure in the manuscript?

Table 1 heading is "Sample types used in the literature reviewed," Where the references to substantiate your claim are. Each of the columns should be adequately referenced.

         Also, remove the row for Abbr. from the table. It can be placed by each sample type that creates a row. E.g. “Stock solution (ss)”

Line 144 – 193: Can be in the discussion or perhaps use figures or illustrations for the methods comparisons

Line 155 - 157: the statement is not clear. Kindly substantiate

Line 247: The starting statement is inappropriate

Table 2: Make this table to be more complex and scientific. It looks scanty but has more references. You can include more columns

Tables 3 & 4: These are good, but the labeling isn’t okay. For Table 3: Instrument type – Description – Advantages – Disadvantages – References

Fig 5: What is the Red on the last wavelength indicating?

Fig 8: should be relabeled and classified as (a) and (b)

Figure 14 (Line 468) and Figure 14 (Line 655). Kindly check this

Line 621: Why is the review limited to data from 1984 – 2023? There is a need to give a detailed explanation of the choice of this long-duration

Conclusion:  Too long. Try to bring up the major point rather than make another discussion at the conclusion part.

Comments on the Quality of English Language

English needs to be rechecked throughout the manuscript to remove grammatical and typological errors

Author Response

Dear Sir or Madam,

thank you very much for your constructive comments, the implementation of which have improved the manuscript. We have highlighted all changes in blue within the manuscript. Please find our responses to your comments below:

1) Abstract: Simplify the abstract by explaining briefly – introduction (2 – 3 lines), aims/objective (1- 2 lines), methods adopted (2-3 lines), trends/outcomes (3 – 4 lines), conclusion (2-3 lines), recommendation/suggestions for further studies (1 – 2lines)

Thank you for that advice. We have completely rewritten the abstract following the suggested structure. We believe this has significantly improved the comprehensibility of our work and will attract more readers.

2) Line 15: Mentions some examples of the LIBS instrument types

We have rewritten the abstract and included the instrument types: “laboratory based, online, portable, and telescopic”.

3) Line 17: which methods are documented?

We have rewritten the abstract and detailed the methods.

4) Line 20 – 21: Unclear statement

We have rewritten the abstract and clarified the statement.

5) Line 22: Include the full meaning of the keyword be an abbreviation

We have included the full meaning in the keywords.

6) Line 102 – 103: Sources of the data for the figure should be clearly stated

As figures 1, 2, 4 – 14 are based on the entire literature reviewed, it is not practical to cite all 153 references. We added a statement “Data based on the literature reviewed”.

7) Line 104: include some statement about methodology in the abstract

We have rewritten the abstract and included a methodology part.

Line 124 – 123: The focus here is different from the objective stated in the abstract (Line 13 - 15). Your work should be consistent and comprehend with each other

We have rewritten the abstract and clarified the objective to make the manuscript more consistent.

8) Line 130 – 131: contracting statement. Kindly clarify

We have revised the paragraph (lines 132 – 139). We hope that this has made the statement clearer.

9) Line 133: Include publication year range

We have added the range in Line 140.

10) Fig. 2: Source and year? Can you explain the necessity of this figure in the manuscript?

As figures 1, 2, 4 – 14 are based on the entire literature reviewed, it is not practical to cite all 153 references. We added a statement “Data based on the literature reviewed”. As explained in the text it shows the hot spots where research on the topic is conducted. For clarification, we added some text passages (lines 228-230).

11) Table 1 heading is "Sample types used in the literature reviewed," Where the references to substantiate your claim are. Each of the columns should be adequately referenced.

Also, Table 1 is based on the entire literature reviewed. We added the same statement as for the figures. Please refer to the additional file for more information.

12) Also, remove the row for Abbr. from the table. It can be placed by each sample type that creates a row. E.g. “Stock solution (ss)”

Thank you for that helpful hint. We deleted the row and added the abbreviations to the first row. Now the table is much clearer.

13) Line 144 – 193: Can be in the discussion or perhaps use figures or illustrations for the methods comparisons

Even if parts of this section appear to be a discussion, only the methodological approach is explained here. This section can therefore unfortunately not be moved to the discussion.  Also, the methods of the literature cited are not described or compared here, but only the approach of the review. This is difficult to visualise.

14) Line 155 - 157: the statement is not clear. Kindly substantiate

We have added an explanation of the IUPAC criterion and improved this paragraph by additional explanations (lines 164 – 169).

15) Line 247: The starting statement is inappropriate

We have deleted the starting statement.

16) Table 2: Make this table to be more complex and scientific. It looks scanty but has more references. You can include more columns

In fact, this table did not look as good as intended. We have improved the table by deleting superfluous lines and removing abbreviations.

17) Tables 3 & 4: These are good, but the labeling isn’t okay. For Table 3: Instrument type – Description – Advantages – Disadvantages – References

We have revised the tables and the proposed changes make the tables look clearer.

18) Fig 5: What is the Red on the last wavelength indicating?

The red lines should show that data has been partially hidden. We have added “(red lines)” to the description.

19) Fig 8: should be relabeled and classified as (a) and (b)

We have relabelled the figure and classified the subfigures as a), b), and c).

20) Figure 14 (Line 468) and Figure 14 (Line 655). Kindly check this

We have corrected that.

22) Line 621: Why is the review limited to data from 1984 – 2023? There is a need to give a detailed explanation of the choice of this long-duration

This long period, in fact the entire time since the beginning of water analysis using LIBS, was chosen to obtain a comprehensive database. The subject is rather niche and it is not possible to include only the last ten years of research in the review and make reliable statements. We have added a statement in lines 151 -152.

Merry Christmas and best regards

Nils Schlatter

Reviewer 4 Report

Comments and Suggestions for Authors

The authors have provided a literature review on using LIBS technique and discussed the instrumentation and elemental analysis. While the review can be a good starting point for the literature check, it is tough to get any details pertaining to the science behind the technique and its strengths and drawbacks vis a vis other techniques. The manuscript does not provide any useful addition to the existing literature in the field.  The authors should discuss the methods in detail to make it more useful.

A few minor comments :

1. Page 1 Line 39 :  Complete the sentence '...sent to a'

2. Page 4 Line 155 :  Give the details of IUPAC convention for the calculation of LoD.

3. Page 8 : It will be useful to provide a small description of each method shown in Fig 3

4. Page 9:  Line 283 How is the surface specific element detection done in the method? How is the surface contribution separated from the bulk in LIBS

5. Page 13 Section 3.7 : It will be useful to provide justification for which elements are hard to identify and what is the way to overcome the issue. It can be useful to plan the usage of LIBS for new analysis.

6. Page 15 Line 451 : Again it's absolutely unclear why the transition metals dont have a clear trend to the investigation via LIBS. I will find a review useful which helps me construct new experiments or simulations based on the reading. It looks more like a heap of information and references.

7. Figure 9 to 13 can be accomodated in a single table rather than histograms.

8.  Finally what I wonder is if it possible to observe the transient reaction intermediates in aqueous solutions using LIBS. 

Comments on the Quality of English Language

minor language checks

Author Response

Dear Sir or Madam,

thank you very much for your constructive comments, the implementation of which have improved the manuscript. We have highlighted all changes in blue within the manuscript. Please find our responses to your comments below:

 1) Page 1 Line 39 :  Complete the sentence '...sent to a'

Thank you for the hint. The word must have disappeared when changing the citation style. We have added “spectrometer” to complete the sentence.

 2) Page 4 Line 155 :  Give the details of IUPAC convention for the calculation of LoD.

We have added an explanation of the IUPAC criterion and improved this paragraph by additional explanations (lines 164 – 169).

3) Page 8 : It will be useful to provide a small description of each method shown in Fig 3

We had hoped that the figures would be self-explanatory, but as you pointed out, it is necessary to provide some additional information. Therefore, we have included brief descriptions of the different methods just before Table 2 and Figure 3 (lines 271 – 311).

4) Page 9:  Line 283 How is the surface specific element detection done in the method? How is the surface contribution separated from the bulk in LIBS

Actually, the difference between “surface” and “inside” analysis is only where the laser is focused. In both analyses one assumes a homogenous solution. The reason for focusing inside the solution rather then on the surface are reduced splashing and cooling of the plasma. We have included brief descriptions of the different methods just before Table 2 and Figure 3 (lines 271 – 311).

5) Page 13 Section 3.7: It will be useful to provide justification for which elements are hard to identify and what is the way to overcome the issue. It can be useful to plan the usage of LIBS for new analysis.

Of course, it is interesting to analyse why some elements are harder to analyse than others. We have tried to discuss this issue in section 4.3 and only show results in 3.7.

6) Page 15 Line 451: Again it's absolutely unclear why the transition metals dont have a clear trend to the investigation via LIBS. I will find a review useful which helps me construct new experiments or simulations based on the reading. It looks more like a heap of information and references.

The transition metals as group of elements show no clear trend compared to the alkali metals as they are a more inhomogeneous group overall than the alkali metals. However, this does not mean that there is a specific trend for certain elements. We added “as a group of elements” to clarify this (lines 503 – 504).

7) Figure 9 to 13 can be accomodated in a single table rather than histograms.

We acknowledge that the figures in the manuscript may blow the manuscript due to the large number of individual graphics. However, we believe that presenting the data in a table may not provide a clear and easily understandable overview. The supplementary file contains a table with all the data, and although we have made efforts to make it easily accessible, it may still require some time to get the same understanding.

8)  Finally, what I wonder is if it possible to observe the transient reaction intermediates in aqueous solutions using LIBS. 

Indeed, that’s a really interesting and multi-layered question, however we did not find any literature about exactly this topic. When using the LSC technique, we observed H lines maybe because the liquid was not completely dried or hydroxides formed. However, we did not investigate this further. While there is extensive research on the distribution of evaporation residue and gas analysis, it is beyond the scope of this manuscript and therefore not included.

Merry Christmas and best regards

Nils Schlatter

Round 2

Reviewer 1 Report

Comments and Suggestions for Authors

I have no further comments on the manuscript. The present version is now ready for the publication.

Reviewer 3 Report

Comments and Suggestions for Authors

The authors have responded satisfactorily to the comments in the revised version. Hence, I recommend the paper for acceptance.

Comments on the Quality of English Language

Minor English checking

Reviewer 4 Report

Comments and Suggestions for Authors

The authors have incorporated the modifications suggested and i will like to recommend the manuscript for publication.